# Coreset for Line-Sets Clustering

**Sagi Lotan**
sagi.lotan@gmail.com

**Ernesto Evgeniy Sanches Shayda**
ernestosanches@gmail.com

**Dan Feldman**
dannyf.post@gmail.com

Robotics & Big Data Labs,
Computer Science Department, University of Haifa

## Abstract

The input to the line-sets $k$-median problem is an integer $k \geq 1$, and a set $\mathcal{L} = \{L_1, \ldots, L_n\}$ that contains $n$ sets of lines in $\mathbb{R}^d$. The goal is to compute a set $C$ of $k$ centers (points in $\mathbb{R}^d$) that minimizes the sum $\sum_{L \in \mathcal{L}} \min_{\ell \in L, c \in C} \operatorname{dist}(\ell, c)$ of Euclidean distances from each set to its closest center, where $\operatorname{dist}(\ell, c) := \min_{x \in \ell} \|x - c\|_2$. An $\varepsilon$-*coreset* for this problem is a weighted subset of sets in $\mathcal{L}$ that approximates this sum up to $1 \pm \varepsilon$ multiplicative factor, for every set $C$ of $k$ centers. We prove that *every* such input set $\mathcal{L}$ has a small $\varepsilon$-coreset, and provide the first coreset construction for this problem and its variants. The coreset consists of $O(\log^2 n)$ weighted line-sets from $\mathcal{L}$, and is constructed in $O(n \log n)$ time for every fixed $d, k \geq 1$ and $\varepsilon \in (0, 1)$. The main technique is based on a novel reduction to a "fair clustering" of colored points to colored centers. We then provide a coreset for this coloring problem, which may be of independent interest. Open source code and experiments are also provided.

## 1 Introduction

In the classic $k$-*mean* clustering problem, the input is a set $P$ of $n$ points in a metric space $(\mathcal{X}, \operatorname{dist})$, and an integer $k \geq 1$. The goal is to compute a set $C^*$ of $k$ centers (points in $\mathcal{X}$) that minimizes the sum of squared distances over each point $p \in P$ to its nearest center in $C^*$, i.e., to compute

$$C^* \in \operatorname*{arg\,min}_{C \subseteq \mathcal{X}, |C| = k} \sum_{p \in P} \tilde{D}(p, C),$$

where $\tilde{D}(p, C) := \min_{c \in C} \operatorname{dist}^2(p, c)$. This problem is arguably the most common clustering problem formulation, both in industry and academy; see references e.g. in [5, 20, 31, 36].

A natural generalization is to replace this input set $P$ of $n$ points by a set $\mathcal{P}$ of $n$ sets in $\mathcal{X}$. The distance from such an input set $P \in \mathcal{P}$ to a set $C^*$ of centers can then be defined as the distance between the closest point-center pair. This problem is called $k$-*mean for sets;* see e.g. [26] and references therein. Its goal is to compute

$$C^* \in \operatorname*{arg\,min}_{C \subseteq \mathcal{X}, |C| = k} \sum_{P \in \mathcal{P}} \tilde{D}(P, C), \tag{1}$$

where $\tilde{D}(P, C) := \min_{p \in P} \tilde{D}(p, C)$.

A special case is where every such set $P \in \mathcal{P}$ is a line $\ell$ in $\mathcal{X} = \mathbb{R}^d$. This problem is called $k$-*mean for lines*; see e.g. [10, 17, 35]. Its goal is to compute a set $C^*$ of $k$ points in $\mathbb{R}^d$ such that

$$C^* \in \operatorname*{arg\,min}_{C \subseteq \mathbb{R}^d, |C| = k} \sum_{\ell \in \mathcal{P}} \tilde{D}(\ell, C). \tag{2}$$

This paper considers a further generalization of the last problem, where the input is a set of $n$ sets, each contains *multiple* lines as follows.

36th Conference on Neural Information Processing Systems (NeurIPS 2022).

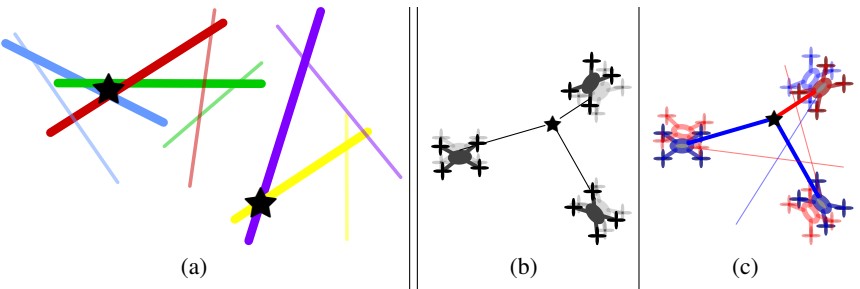

(a)    (b)    (c)

Figure 1: **Line-sets Clustering:***(a) The $k = 2$ line-sets mean (black stars) of $n = 5$ sets (in different colors), each contains a pair ($m = 2$) of lines (same color) in the plane ($\mathcal{X} = \mathbb{R}^2$). The closest line to the stars in each pair of lines is bold.* **Application:** *(b) A set of $n = 3$ drones, each equipped with an on-board 2D camera, takes a snapshot of a point in $\mathbb{R}^3$ (the black $k = 1$ star). Due to the missing depth, each such point corresponds to a pixel in the 2D image's plane, or a line in $\mathbb{R}^3$ which is the intersection of the image's plane with the camera's pin-hole. (c) The estimated position of the drone from the noisy observations is inaccurate, so we use $m = 2$ guesses or consecutive images over time (blue and red line). One of them (in bold) is closer to the observed point.*

**Line-sets clustering.**   The input to the line-sets $k$-mean problem is an integer $k \geq 1$, and a set $\mathcal{L} = \{L_1, \ldots, L_n\}$ of $n$ sets, where each set $L = \{\ell_1, \ldots, \ell_m\}$ in $\mathcal{L}$ consists of $m \geq 1$ lines in $\mathbb{R}^d$. The goal is to compute a set $C^*$ of $k$ points in $\mathbb{R}^d$ that minimizes its sum of squared distances over every set of lines in $\mathcal{L}$, i.e.,

$$C^* \in \underset{C \subseteq \mathbb{R}^d, |C| = k}{\arg\min} \sum_{L \in \mathcal{L}} \tilde{D}(L, C). \tag{3}$$

Here, $\tilde{D}(L, C) := \min_{\ell \in L} \tilde{D}(\ell, C)$ is the shortest squared distance between a point in $C$ to its closest line in $L$; see Fig 1.

## 1.1   Motivation

The classic $k$-clustering of points has numerous applications, and the $k$ line-sets clustering is its natural generalization. It is thus not surprising that these applications can be easily generalized and extended. We give examples from different research fields as follows.

**Handling missing data.** A natural application is clustering of a data with missing values, as described e.g. in [10]. Here, each record in the input data-set is represented by a point in $d$ dimensions. If an entry is missing, we may replace it by all possible values, resulting in a line in $\mathbb{R}^d$ that is parallel to one of the axes. Similarly, if the missing entry is discrete, i.e., can be one in a finite set of options, it results in a set of points. Our paper handles combination of them both. An example of this can be found in Dataset 4 in our experimental results for California housing. The entries may be the number of the apartment or floor, which corresponds to a set of values. A monthly rent is a continuous variable, which corresponds to a line. Together they correspond to line-sets. Another perspective would be of sets clustering with missing values, where most of the applications in [26] may be generalized using our work.

**Computer vision.** In the fundamental 3D model reconstruction problem, we are given $n$ pixels (features). Each pixel is from a different 2D image that captures the same point in $\mathbb{R}^3$, i.e., the real world. Such a 2D pixel corresponds to a line in $\mathbb{R}^3$ that passes through the camera's pinhole and the pixel in the image plane. The goal is to estimate the location of the point in $\mathbb{R}^3$, based on the set of $n$ 2D images. The task of estimating a 3D point location given a single set of lines is called *Triangulation*. Due to uncertainty and noise, feature extraction algorithms usually identify $m \geq 1$ pixels in each image as the captured point, without certainty about which of the pixels are noise and which are correct. Here, the center is this desired point in $\mathbb{R}^3$, and every image is represented by $m$ lines. To obtain a model or a "3D point-cloud" [18, 22] we may wish to compute $k$ points (centers) of the 3D object. See Fig. 1 (right).

**Colored point-sets clustering applications.** This paper reduces the line-sets clustering problem to clustering sets of colored points, which has its own applications. An example is the facility location problem, as in the home and work example in Section 3. Another application is text documents clustering, as explained in Section 5. Here, every document is represented by a set of colored vectors, where each vector corresponds to a paragraph in a "bag of words" representation. Each color represents a different type of paragraph, e.g., introduction, summary, main section, theorem, etc.

More generally, the problem is relevant where each facility is useful for a specific subset of input clients.

We hope that this paper will open the research toward more complex and general shapes that may yield better approximations to high-ways and polygons such as $k$ segments (as in [38, 25, 10]), arcs or $k$-subspaces instead of lines.

## 1.2 Generalizations and computation models

Our main technique to handle the problems above, and especially their streaming, distributed, and parallel versions, is to design data summaries (often called coresets) for the $k$ line-sets clustering problem, and its corresponding set of queries. We may then run naive algorithms such as exhaustive search on the small coreset to extract the approximated solution, or apply existing heuristics and obtain faster results.

In the following definition, a pseudo distance $\tilde{D}(p, q)$ assigns a non-negative number for any pair $p, q \in R^d$. The pseudo distance between sets is then the distance between the closest pair, as in (1) above. See exact definition and constraints in Definition 2.1.

**Definition 1.1** (Coreset). *Let $k \geq 1$ be an integer, and $\mathcal{L} = \{L_1, \ldots, L_n\}$ be an input set, where each $L = \{\ell_1, \ldots, \ell_m\} \in \mathcal{L}$ consists of $m \geq 1$ lines in $\mathbb{R}^d$. For a given approximation error parameter $\varepsilon \in (0, 1)$ and an integer $k \geq 1$, an $\varepsilon$-coreset of $\mathcal{L}$ for $k$ line-sets clustering over a pseudo distance $\tilde{D}$ is a weighted set $\mathcal{S} \subseteq \mathcal{L}$ that approximates the sum of pseudo distances to $\mathcal{L}$ from every set $C$ of $k$ centers in $\mathbb{R}^d$, up to a multiplicative factor of $1 \pm \varepsilon$. More precisely, there is a weight function $v : \mathcal{S} \to [0, \infty)$ such that for every set $C \subseteq \mathbb{R}^d$ of size $|C| \leq k$,*

$$\left| \sum_{L \in \mathcal{L}} \tilde{D}(L, C) - \sum_{L \in \mathcal{S}} v(L)\tilde{D}(L, C) \right| \leq \varepsilon \sum_{L \in \mathcal{L}} \tilde{D}(L, C).$$

In particular, an (optimal) line-sets $k$-mean $C^*$ of the weighted set $(\mathcal{S}, v)$ is a $(1 + \varepsilon)$-approximation to the $k$-line-sets mean of $\mathcal{L}$. This is the strongest type of coresets (sometimes called "strong coreset" [33, 41]), in the sense that it approximates *every* query $C$, and not just, say, the optimal solution. It is also a weighted subset of the input, unlike a sketch, linear combination, or general subsets of lines in $\mathbb{R}^d$ which are not a subset of $\mathcal{L}$; see next Section 1.3.

## 1.3 Related work

Clustering $n$ points by $k$ center points is a fundamental problem in machine learning [40]. Applications can be found in operations research [6, 8], statistics [20] and computational geometry [4, 28, 43], including constant factor approximations (randomized [2] and deterministic [7]). Many of the recent results [13, 21] are based on coresets (under different definitions) that can usually be computed in time that is near-linear in the input size and number $k$ of centers. The size of these coresets is usually $(k/\varepsilon)^{O(1)}$, where $\varepsilon \in (0, 1)$ is a parameter that represents the approximation error.

For the special case $m = 1$, our coreset in Definition 1.1 is for $k$-means of $n$ lines. Such a coreset was suggested in [35] whose size is $\log^2(n)k^{o(k)} \cdot \varepsilon^{-2}$, for any constant $\varepsilon > 0$, and is improved in our paper. For the case that the lines are parallel to the axes, coresets were suggested in [10]. These results also hold for $j$-dimensional linear subspaces, but only if they are parallel to the axes of $\mathbb{R}^d$. The main motivation is handling $j$ missing entries in each database's record (point).

Coresets for input sets of points, lines, and fair clustering appeared relatively recently. Coreset for (point)-sets clustering as in Eq. (1), where each set consists of $m$ points, was suggested in [26] and its size is $\left(\frac{\log n}{\varepsilon}\right)^2 k^{O(m)}$.

Our main technical result is a coreset for colored (points) clustering, where the sets and centers are colored, and each input point can be served by a center of the same color. This approach is strongly related to fair clustering, where each group is represented by a color.

Variations of fair clustering were suggested in [12, 23, 27] and references therein. Coresets for fair clustering were suggested in [39]. This paper generalizes some of these results in the sense that the input points are colored sets, but the centers are also colored.

Our coreset constructions, as many of the recent coreset constructions, is based on the Feldman-Langberg framework [9, 11, 15, 29]. This framework reduces the problem of computing a coreset – to the problem of computing the importance (sensitivity) for each of the $n$ input points/sets.

## 1.4 Main contributions

We suggest the first coreset construction for line-sets clustering whose size (number of weighted sets) is sub-linear in the number $n$ of input sets. This is by a reduction to another problem, fair clustering (colored points) with colored centers, which may be of independent interest. We then suggest a coreset for this type of fair clustering problems. More precisely, we provide the following contributions for every pseudo distance $\tilde{D}$, constant approximation error $\varepsilon \in (0,1)$, and constant integers $d, k, m \geq 1$. See the corresponding theorems for exact bounds.

($i$) An $\varepsilon$-coreset $S \subseteq \mathcal{L}$ for line-sets clustering of any set $\mathcal{L}$ of $n$ sets , each consists of at most $m$ lines; see Theorem 4.1

($ii$) An $\varepsilon$-coreset $S \subseteq \mathcal{P}$ for colored sets clustering of any set $\mathcal{P}$ of $n$ sets in $\mathcal{X}$; See Theorem 4.2

Each of these coresets has size $|\mathcal{S}| \in O(\log^2 n)$ and can be computed in $O(n \log n)$ time, with high probability. The following results are straightforward implication of the above results, using existing merge-reduce frameworks [1]:

($iii$) Support for streaming data in insertion time and memory that is poly-logarithmic in the number $n$ of the sets seen so far in the stream.

($iv$) Support for parallel computations on data that may be distributed on $M$ machines using $1/M$ amortized insertion time.

($v$) Support for deletion of sets from the stream in poly-logarithmic time during the streaming, but using memory that is linear with $n$.

($vi$) FPTAS in time $O(n \log n)$, by running an exhaustive search on the corresponding small corsets.

**Dependencies on input parameters.** Our first result above is in fact an LTAS (linear time approximation scheme), since the asymptotic running time depends polynomially on $1/\varepsilon$, and near-linear in $n$. The dependency on $d$ is polynomial, and the exponential dependency on $k$ and $m$ is unavoidable due to known lower bounds for special cases e.g. in [19, 26]. These are worst-case bounds, and the actual approximations errors are much smaller in practice as shown in the experimental results; see Section 5.

## 1.5 Novel technique: from line-sets to fair clustering

Existing coresets for point-sets clustering are heavily based on the triangle inequality between pair of points, which is not satisfied for the case of lines. On the other hand, existing coresets for lines were reduced to coresets for weighted centers, which do not support sets as an input. The main challenge in this paper is to combine these two results. This is by suggesting a novel reduction and coresets for colored-weighted centers, which may be of independent interest and borrows techniques from fair clustering. To our knowledge, this is the first link of this type between geometric shapes and fairness. We hope it will open the door for many other coresets; see Section 6.

Our reduction is based on the following three steps.

**Grouping.** Algorithm 3 in Section 3.2 recursively chooses a very "dense" constant fraction of sets $\mathcal{L}^m$ that are close to $m$ center points (robust medians) $\mathcal{B}^m = \{b_1, \ldots, b_m\}$ ; see Definition 2.5. The $i$th line in each of these sets is then translated to its median $b_i$, for every $i \in [m]$. At this stage we reduced our problem to compute sensitivity for $|\mathcal{L}^m| \in O(n)$ sets of lines that form $m$ "stars"; see Fig. 2(a).

**Reduction to points.** The distance from each line that intersects a robust median $b_i$ (say, the origin) to a query center $c_i \in C$ is its distance to a unit vector in the same direction, weighted by the norm of the vector; see Fig. 2(b). This distance can be approximated, up to a constant factor, by replacing $c_i$ with its projection on the sphere $c_i / \|c_i\|$ and its antipodal point $-c_i / \|c_i\|$.

**Colored sets and centers.** The new problem is now to compute sensitivity for $n$ sets, each contains $m$ *colored points* on $m$ unit spheres; see Fig. 2(c). Each center in the set $C$ of $k$ queries is duplicated $m$ times in $m$ colors. The result is a set of $2mk$ weighted centers on $m$ unit sphere with an additional constraint: the $i$th center can "serve" only the lines on the $i$th sphere.

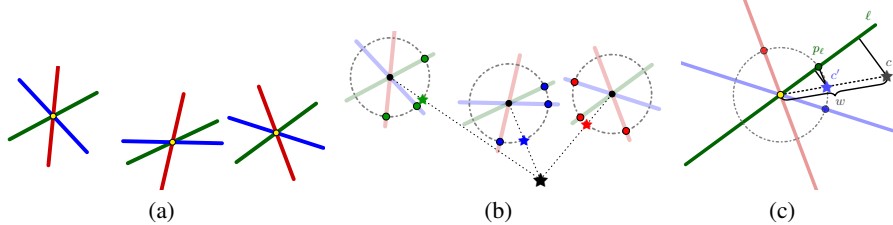

(a)                    (b)                   (c)

Figure 2: **(a)** *A set $\mathcal{L}$ of 3 triplets of lines, each in a different color. The first line in each triplet intersects the same yellow (robust median) point $b_1$, and similarly for the other lines and medians $b_2$, $b_3$.* **(b)** *The distance from every colored line to a query center $c \in \mathbb{R}^2$ (black star) is the same as its distance from the star in the same color multiplied by the distance $w$ from the center to its median.* **(c)** *The distance between a line $\ell$ (in green) and the point $c$ (black) is the distance between $c'$ (blue) and $\ell$ multiplied by $w$ (black brackets). This distance is approximated by the distance $\|c' - p_\ell\|$ as stated in Claim C.3.*

## 2 Preliminaries

The results in this paper hold not only for squared distance functions but for other functions such as non-squared distances or M-estimators. To generalize this notion, we use the following pseudo distance over a metric space $(\mathcal{X}, \mathrm{dist})$; see many examples of such functions in [26]. From Section 3.2, i.e., for the case of input lines, we assume that $\mathcal{X} = \mathbb{R}^d$ and $\mathrm{dist}$ is the Euclidean norm. That is, the metric space is $(\mathbb{R}^d, ||.||_2)$ but lip below can still be any function that satisfies the following log-log Lipschitz condition.

**Definition 2.1** (Pseudo-distance $\tilde{D}$). *Let* $\mathrm{lip} : [0, \infty) \to [0, \infty)$ *be a non-decreasing function that satisfies the following condition: There is a constant $r \in (0, \infty)$ such that for every $x, z > 0$ we have* $\mathrm{lip}(zx) \leq z^r \mathrm{lip}(x)$. *Let $(\mathcal{X}, \mathrm{dist})$ be a metric space, and $\tilde{D}$ be a function that maps every pair of points $p, c \in \mathcal{X}$ to*

$$\tilde{D}(p, c) := \mathrm{lip}(\mathrm{dist}(p, c)).$$

*For a pair of finite sets $P, C \subseteq \mathcal{X}$, denote $\tilde{D}(P, C) := \min_{p \in P, c \in C} \tilde{D}(p, c)$, $\tilde{D}(P, c) := \tilde{D}(P, \{c\})$ and $\tilde{D}(p, C) = \tilde{D}(\{p\}, C)$.*

The motivation behind the last definition is that it satisfies the following pair of properties.

**Lemma 2.2** (weak triangle [42]). *Let $\mathcal{X}$, $\tilde{D}$, and $r$ be as defined in Definition 2.1. Then $\tilde{D}$ satisfies the following ("weak triangle") inequalities (i)–(ii) for every $p, q, c \in \mathcal{X}$:*

*(i)* *For $\rho = \max\{2^{r-1}, 1\}$,* $\quad \tilde{D}(p, q) \leq \rho(\tilde{D}(p, c) + \tilde{D}(c, q))$.

*(ii)* *For $\phi = (4r - 4)^{r-1}$,* $\quad \tilde{D}(p, c) - \tilde{D}(q, c) \leq \phi \tilde{D}(p, q) + \frac{\tilde{D}(p, c)}{4}$.

In the rest of the paper, for an integer $n \geq 1$ we denote $[n] = \{1, \ldots, n\}$. Also, unless otherwise stated $\tilde{D}, \mathcal{X}, r$ are as stated in Definition 2.1.

For both the line-sets clustering and the colored-sets clustering problems/coresets, the input is a set of $n$ sets, each of size $m$. We call it $(n, m)$-set for short as follows.

**Definition 2.3** ($(n, m)$-set). *For a given integer $m \geq 1$, an $m$-set $P$ is a set of $m$ items, i.e. $|P| = m$. For an additional integer $n \geq 1$, an $(n, m)$-set is a set of $n$ $m$-sets.*

**Definition 2.4** ($(n, m)$-ordered-set). *An $m$-ordered set $P$ is an ordered-set of $m \geq 1$ points in $\mathcal{X}$. An $(n, m)$-ordered-set is a set of $n \geq 1$ $m$-ordered-sets.*

Informally, a robust median for an optimization problem at hand is an element $b$ that approximates the optimal value of this optimization problem, with some leeway on the number of input elements considered. In the context of facility location, the facility (center) needs to serve only a subset of the closest clients (input points).

Let $\mathcal{P}$ be an $(n, m)$-set, $C \subseteq \mathcal{X}$ and $\gamma \in (0, 1]$. We denote by $\mathrm{closest}(\mathcal{P}, C, \gamma)$ the set that is the union of $\lceil \gamma|\mathcal{P}| \rceil$ sets $P \in \mathcal{P}$ with the smallest value of $\tilde{D}(P, C)$, i.e.,

$$\mathrm{closest}(\mathcal{P}, C, \gamma) \in \operatorname*{arg\,min}_{\mathcal{Q} \subseteq \mathcal{P}: |\mathcal{Q}| = \lceil \gamma|\mathcal{P}| \rceil} \sum_{P \in \mathcal{Q}} \tilde{D}(P, C). \tag{4}$$

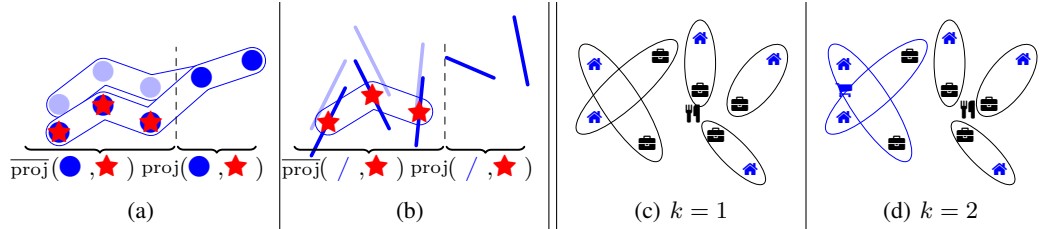

Figure 3: *Example sets projections for Definition 2.7:* **(a)** *The projection of* $P = \bullet$ *(top 5 points) onto the ordered set* $\mathcal{B} = \star$ *(red stars) yields* $T(P, \mathcal{B}) = \bullet$ *(5 dark blue points), i.e., the union of 3 points* $\mathrm{proj}(P, \mathcal{B})$ *(3 on the stars) with the* 2 *remaining points* $\overline{\mathrm{proj}}(P, \mathcal{B})$. **(b)** *The projection of* $L = /$ *(top 5 lines) onto the ordered set* $\mathcal{B}$ *(red stars) yields* $T(L, \mathcal{B})$ *(5 dark blue lines), i.e., the union of 3 lines* $\mathrm{proj}(P, \mathcal{B})$ *(translated to to the stars) with the* 2 *remaining lines* $\overline{\mathrm{proj}}(L, \mathcal{B})$. **Colored-set** $k$**-clustering:** *The input is a set of* $n = 5$ *pairs of points that correspond to home/work addresses (in blue and black).* **(c)** $k = 1$ *center, which is a restaurant (black fork and knife). Its distance to a person is the distance to her office (black suitcase).* **(d)** $k = 2$ *centers: a restaurant (black fork and knife) and a grocery shop (blue cart). Each person chooses the closest restaurant to her office or the closest grocery to her home.*

For simplicity of notation, we define $\mathrm{closest}(\mathcal{P}, C) := \mathrm{closest}(\mathcal{P}, C, \frac{1}{|\mathcal{P}|})$ as one of the closest items to $C$ in $\mathcal{P}$. Here and in the rest of the paper ties are broken arbitrarily.

**Definition 2.5** (Robust median [16])**.** *Let* $\mathcal{P}$ *be an* $(n, m)$*-set in* $\mathcal{X}$, $\gamma \in (0, 1]$, *and*
$$\tilde{D}^*(\mathcal{P}, \gamma) = \min_{b \in \mathcal{X}} \sum_{P \in \mathrm{closest}(\mathcal{P}, \{b\}, \gamma)} \tilde{D}(P, b).$$
*For* $\tau \in (0, 1)$ *and* $\alpha \geq 0$, *a point* $b \in \mathcal{X}$ *is a* $(\gamma, \tau, \alpha)$*-median for* $\mathcal{P}$ *if*
$$\sum_{P \in \mathrm{closest}(\mathcal{P}, \{b\}, (1-\tau)\gamma)} \tilde{D}(P, b) \leq \alpha \cdot \tilde{D}^*(\mathcal{P}, \gamma).$$

**Definition 2.6** (Translation)**.** *Let* $b$ *be a point in* $\mathcal{X}$. *Than* $(i)$ *For every* $p \in \mathcal{X}$, *we define* $T(p, b) := b$, *and* $(ii)$ *If* $\mathcal{X} = \mathbb{R}^d$, *for every line* $\ell$ *in* $\mathbb{R}^d$ *we define* $T(\ell, b)$ *to be a line parallel to* $\ell$ *that intersects* $b$.

In what follows, we define the projection of a set (points or lines) over a set of points; see Fig. 3.

**Definition 2.7** (Set projection)**.** *Let* $m, j$ *be a pair of integers such that* $1 \leq j \leq m$. *Let* $P$ *be either an* $m$*-set of points in* $\mathcal{X}$ *or an* $m$*-set of lines in* $\mathcal{X} = \mathbb{R}^d$. *Let* $\mathcal{B} = (b_1, \ldots, b_j)$ *be an ordered* $j$*-set of points in* $\mathcal{X}$. *Let* $p_1 \in P$ *denote the closest item* $\{p_1\} = \mathrm{closest}(P, \{b_1\})$ *to* $b_1$. *For every integer* $i \in \{2, \ldots, j\}$, *recursively define* $p_i$ *to be the closest item* $\{p_i\} = \mathrm{closest}(P \setminus \{p_1, \ldots, p_{i-1}\}, \{b_i\})$ *in* $P \setminus \{p_1, \ldots, p_{i-1}\}$ *to* $b_i$. *We denote,*

    *(i)* $\mathrm{proj}(P, \mathcal{B}) := \{T(p_1, b_1), \ldots, T(p_j, b_j)\}$ *as the* $j$ *items in* $P$ *that were projected onto* $\mathcal{B}$.

    *(ii)* $\overline{\mathrm{proj}}(P, \mathcal{B}) := P \setminus \{p_1, \ldots, p_j\}$ *as the* $m - j$ *items in* $P$ *that were not projected onto* $\mathcal{B}$.

    *(iii)* $T(P, \mathcal{B}) := \mathrm{proj}(P, \mathcal{B}) \cup \overline{\mathrm{proj}}(P, \mathcal{B})$ *as the* projection *of* $P$ *onto* $\mathcal{B}$.

    *(iv) For projection over an empty set, we define* $\overline{\mathrm{proj}}(P, \emptyset) := T(P, \emptyset) := P$.

## 3 Sensitivity

Our main coreset construction (Algorithm 4) is a standard non-uniform sampling algorithm based on Feldman-Langberg framework [14]. The main challenge in this framework is to compute the importance of each input set, known as sensitivity. Our main contribution is an algorithm that computes this sensitivity with provable guarantees on its running time and the total sensitivity, which implies the size of the resulting coreset.

As explained in the novelty section, we provide coreset construction for line-sets, by bounding sensitivity of a constant fraction of the input lines recursively. Section 3.2 provides sensitivity bound for lines-sets clustering, by translating the input lines into robust medians, converting the translated lines into colored points, and then compute sensitives for colored-sets as in Section 3.1.

**Definition 3.1** (Line-set Sensitivity)**.** *Let* $\mathcal{L}$ *be an* $(n, m)$*-set of lines in* $\mathbb{R}^d$, *and let* $k \geq 1$ *be an integer. We define the* sensitivity *of every such set* $L \in \mathcal{L}$ *of lines to be*
$$S_{\mathcal{L}, k}(L) = \sup_{C \subseteq \mathbb{R}^d, |C| = k} \frac{\tilde{D}(L, C)}{\sum_{L' \in \mathcal{L}} \tilde{D}(L', C)},$$
*where the supremum is over every set* $C$ *of* $k$ *points in* $\mathbb{R}^d$ *such that* $\tilde{D}(L, C) > 0$.

## 3.1 Colored sets

In this subsection, we bound the sensitivity for the *colored sets clustering*. Here, the input is a set $\mathcal{P}$ of $n$ $m$-ordered-sets (the order resembles the colors of the points) in a metric space $(\mathcal{X}, d)$ equipped with a distance function $\tilde{D}$, and an integer $k \geq 1$. The goal is to compute a colored set $\subseteq \mathcal{X} \times [m]$ of $k$ pairs, each consisting of a point and a index (color) in $[m]$ that minimizes the sum of distances to the sets. Here, the distance to each ordered-set $P \in \mathcal{P}$ is the minimum distance from a center in to a point in $P$ which has the same color,

$$\in \underset{C' \subseteq \mathcal{X} \times [m], |C'| = k}{\arg \min} \sum_{(p_1, \ldots, p_m) \in \mathcal{P}} \min_{(c,t) \in C'} D(c, p_t).$$

**Example.** Suppose we want to provide a convenient food source for $n$ workers whose home and work addresses are represented by $n$ pairs of GPS coordinates (points on the plane). Each person can either buy ingredients at the grocery and make a lunch box at home, or go to a restaurant during lunchtime; see Fig 3.

In order to make the reduction to sets of lines, we need coreset for this problem that supports weights. Hence, we define the weighted version of this problem.

**Definition 3.2** (weighted colored center). *A weighted colored center in $\mathcal{X}$ is a triplet $(c, w, t)$, where $c \in \mathcal{X}$, $w \geq 0$, and $t \geq 1$ is an integer. We define the weighted distance from an $m$-ordered set $P = (p_1, \ldots, p_m)$ to a colored weighted center $c' = (c, w, t)$ to be*

$$\tilde{D}(P, c') = \tilde{D}(P, (c, w, t)) = \begin{cases} w \cdot \tilde{D}(p_t, c), & \text{if } t \leq m \\ \infty, & \text{otherwise} \end{cases}.$$

*The distance between a finite set $C$ of weighted colored centers, and an $m$-ordered-set $P$ is defined by $\tilde{D}(P, C) = \min_{c' \in C} \tilde{D}(P, c')$, and the cost between $C$ and an $(n, m)$-ordered set $\mathcal{P}$ is the sum $\tilde{D}(\mathcal{P}, C) = \sum_{P \in \mathcal{P}} \tilde{D}(P, C)$.*

**Overview of Algorithm 1.** Given a set $\mathcal{P}$ of $m$-ordered-sets and an integer $k \geq 1$, Algorithm 1 computes a set $\mathcal{B}_k^m$ of center points (used only for the analyses) and a set $\mathcal{P}_k^m \subseteq \mathcal{P}$ of "similar" $m$-ordered-sets, which are approximately equally important for the problem at hand; see Lemma 3.3. On the $\ell$-th iteration of the external loop (Line 2), the algorithm computes a fraction of $O(\frac{1}{k^m})$ $m$-sets from $\mathcal{P}_{\ell-1}^m$ that are similar in the sense that there are $m$ dense balls of small radii, each contains at least one point from each set.

---

**Algorithm 1:** CS-DENSE$(\mathcal{P}, k)$

**Input** : An $(n, m)$-ordered-set $\mathcal{P}$ in $\mathcal{X}$, and an integer $k \geq 1$.
**Output** : A pair $(\mathcal{P}_k^m, \mathcal{B}_k^m)$, where $\mathcal{P}_k^m \subseteq \mathcal{P}$ and $\mathcal{B}_k^m \subseteq \mathcal{X} \times [m]$ ; see Lemma 3.3.

1 $\mathcal{P}_1^0 := \mathcal{P}$ ; $\mathcal{B}_1^0 := \emptyset$; $\tau := \dfrac{1}{20}$

2 **for** $r := 1$ **to** $k$ **do**

3      **for** $\ell := 1$ **to** $m$ **do**

4          $\overline{\mathcal{P}}_r^{\ell-1} := \left\{ \overline{\text{proj}}(P, \mathcal{B}_r^{\ell-1}) \big| P \in \mathcal{P}_r^{\ell-1} \right\}$ // see Definition 2.7

5          Compute a $\left( \dfrac{1}{2k}, \tau, 2 \right)$-median $b_r^\ell \in \mathcal{X} \times [m - (\ell - 1)]$ for $\overline{\mathcal{P}}_r^{\ell-1}$

         // see Definition 2.5, and suggested implementation in Algorithm 6.

6          $\mathcal{P}_r^\ell := \left\{ P \in \mathcal{P}_r^{\ell-1} \ \middle| \ \overline{\text{proj}}(P, \mathcal{B}_r^{\ell-1}) \in \text{closest} \left( \overline{\mathcal{P}}_r^{\ell-1}, \{b_r^\ell\}, \frac{1-\tau}{4k} \right) \right\}$

         // $\mathcal{P}_r^\ell$ contains every $m$-set $P$ such that $\overline{\text{proj}}(P, \mathcal{B}_r^{\ell-1})$ is in the fraction of
            the closest $(1 - \tau)/(4k)$ sets in $\overline{\mathcal{P}}^{\ell-1}$ to the center $b_r^\ell$; see
            Definition 2.7.

7          $\mathcal{B}_r^\ell := \mathcal{B}_r^{\ell-1} \bigcup \{b_r^\ell\}$

8      $\mathcal{P}_{r+1}^0 := \mathcal{P}_r^m$ ; $\mathcal{B}_{r+1}^0 := \mathcal{B}_r^m$

9 **Return** $(\mathcal{P}_k^m, \mathcal{B}_k^m)$

---

**Lemma 3.3** (sensitivity of colored-sets). *Let $\mathcal{P}$ be an $(n, m)$-ordered-set in $\mathcal{X}$, $k \geq 1$ be an integer, and let $(\mathcal{P}_k^m, \mathcal{B}_k^m)$ be the output of a call to CS-DENSE$(\mathcal{P}, k)$; see Algorithm 1. Then $|\mathcal{P}_k^m| \in \Theta(n)$, and for every set $P \in \mathcal{P}_k^m$ and a set $C \subseteq \mathcal{X} \times [0, \infty) \times [m]$ of $|C| = k$ weighted colored centers such that $\tilde{D}(P, C) > 0$, we have $\dfrac{\tilde{D}(P, C)}{\sum_{P' \in \mathcal{P}} \tilde{D}(P', C)} \leq \dfrac{2k}{|\mathcal{P}_k^m|}$.*

## 3.2 Line-sets

**Overview of Algorithm 2.** Given an $(m, m)$-set $\mathcal{L}$ of lines, and a set $B \subseteq \mathbb{R}^d$ that both satisfy the condition of Lemma C.4, and an integer $k \geq 1$. Algorithm 2 outputs a function $s : \mathcal{L} \to [0, \infty)$ that maps every set $L \in \mathcal{L}$ of lines to an upper bound $s(L)$ of its sensitivity. This is done by a reduction to colored-sets clustering. Intuitively, the distance between a projected set of lines (see Fig. 2 (a)) and a center can be approximated by a distance between a colored set and a set of colored centers (projections of the original center) as shown in Fig. 2(b)

---

**Algorithm 2:** GROUPED-SENSITIVITY$(\mathcal{L}, B, k)$

**Input** : An integer $k \geq 1$, an $(n, m)$-set $\mathcal{L}$ of lines, and a set $B = (b_1, \ldots, b_m)$ of $m$ points both in $\mathbb{R}^d$ such that for every set $L = (\ell_1, \ldots, \ell_m) \in \mathcal{L}$ and every $i \in [m]$, the line $\ell_i$ intersects $b_i$.

**Output** : A sensitivity function $s : \mathcal{L} \to [0, \infty)$, for every set $L \in \mathcal{L}$.

1 For every $i \in [m]$, set $\mathbb{S}_i := \left\{ x \in \mathbb{R}^d \mid \|x - b_i\| = 1 \right\}$

2 For every $L \in \mathcal{L}$, set $P(L) := (\ell_i \cap \mathbb{S}_i)_{i=1}^m$ // A $2m$-ordered set

3 $\mathcal{P} := \left\{ P(L) \mid L \in \mathcal{L} \right\}$// An $(n, 2m)$-ordered set

4 **while** $|\mathcal{P}| > 2mk$ **do**

5     $(\mathcal{P}_k^m, \mathcal{B}_k^m) := $ CS-DENSE$(\mathcal{P}, mk)$// see Algorithm 1

6     **for** *every* $P \in \mathcal{P}^m$ **do**

7        $s'(P) := \dfrac{2mk}{|\mathcal{P}_k^m|}$// see Lemma 3.3

8     $\mathcal{P} := \mathcal{P} \backslash \mathcal{P}_k^m$

9 **for** *every* $Q \in \mathcal{P}$ **do**

10     $s'(Q) := 1$

11 **for** every $L \in \mathcal{L}$ **do** $s(L) := \sqrt{2} s'(P(L))$.

12 **Return** $s$

---

**Overview of Algorithm 3** Given an $(n, m)$-set $\mathcal{L}$ of lines, and an integer $k \geq 1$, Algorithm 3 outputs a pair of sets $(\mathcal{L}^{m+1}, \mathcal{B}^m)$. All the $m$-sets in $\mathcal{L}^{m+1}$ are similar in the sense that their sensitivity with respect to original problem is small and equal. The algorithm consists of two parts. The first part (Lines 3-8), extracts a subset of the original data that can be "grouped" (see Fig. 2(a)) with little effect on the sensitivity related to the original set. The second part (Lines 9-10), uses Algorithm 2 to extract the biggest subset of this subset whose sensitivity can be bounded.

---

**Algorithm 3:** LS-DENSE$(\mathcal{L}, k)$

**Input** : An $(n, m)$-set $\mathcal{L}$ and an integer $k \geq 1$.

**Output** : A pair $(\mathcal{L}^m, \mathcal{B}^m)$, where $\mathcal{L}^m \subseteq \mathcal{L}$ and $\mathcal{B}^m \subseteq \mathbb{R}^d$ is an ordered set.

1 $\tau := \frac{1}{20}$

2 $\bar{\mathcal{L}}^0 := \mathcal{L}^0 := \mathcal{L}$ ; $\mathcal{B}^0 := \emptyset$

3 **for** $i := 1$ **to** $m$ **do**

4     Compute a $\left( \dfrac{1}{2k}, \tau, 4 \right)$-median $b^i \in \mathbb{R}^d$ for $\bar{\mathcal{L}}^{i-1}$
    // see Algorithm 6 for suggested implementation

5     $\mathcal{L}^i := \left\{ L \in \mathcal{L}^{i-1} \mid \overline{\text{proj}}(L, \mathcal{B}^{i-1}) \in \text{closest}\left( \bar{\mathcal{L}}^{i-1}, \{b^i\}, \frac{1-\tau}{4k} \right) \right\}$

6     $\mathcal{B}^i := \mathcal{B}^{i-1} \bigcup \{b^i\}$

7     $\bar{\mathcal{L}}^{i-1} := \left\{ \overline{\text{proj}}(L, \mathcal{B}^{i-1}) \mid L \in \mathcal{L}^{i-1} \right\}$// see Fig 3

8 $\mathcal{L}' := \left\{ \text{proj}(L, \mathcal{B}^m) \mid L \in \mathcal{L}^m \right\}$

9 $s := $ GROUPED-SENSITIVITY$(\mathcal{L}', \mathcal{B}^m, k)$// see Algorithm 2

10 $\mathcal{L}^{m+1} := \underset{L \in \mathcal{L}^m}{\arg\min} \; s\left( \text{proj}(L, \mathcal{B}^m) \right)$// $\mathcal{L}^{m+1}$ is a set of sets
    // biggest cluster whose sets have equal sensitivity

11 **Return** $(\mathcal{L}^{m+1}, \mathcal{B}^m)$

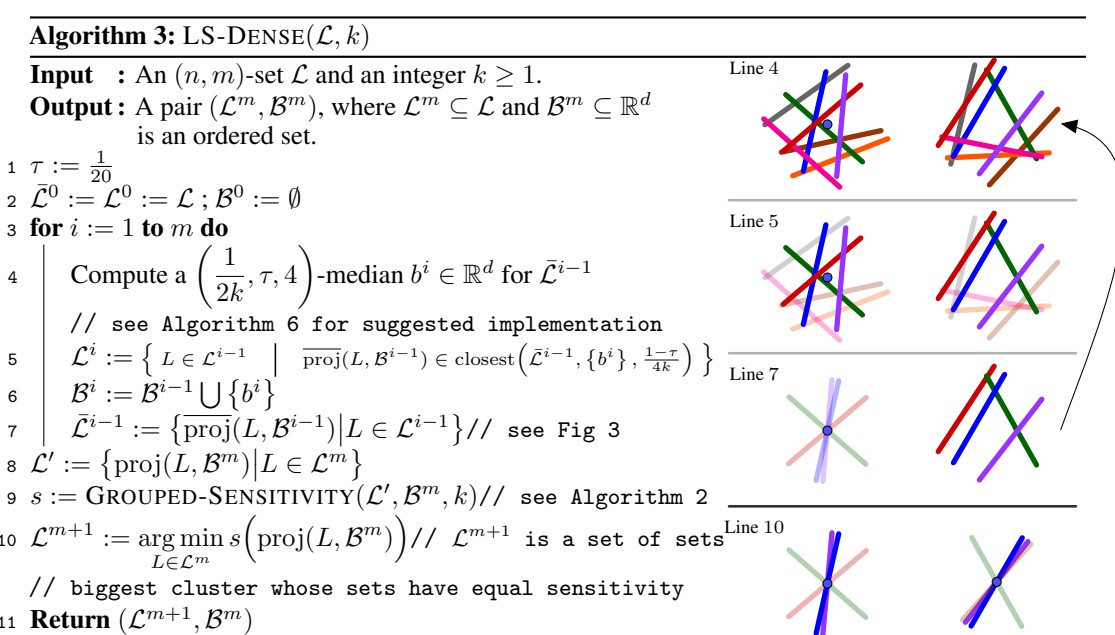

---

**Lemma 3.4.** *Let $\mathcal{L}$ be an $(n, m)$-set of lines in $\mathbb{R}^d$, and $k \geq 1$ be an integer. Let $(\mathcal{L}^{m+1}, \mathcal{B}^m)$ be the output of a call to* LS-DENSE$(\mathcal{L}, k)$; *see Algorithm 3. Then, for every $L \in \mathcal{L}^{m+1}$, we have* $S_{\mathcal{L}, k}(L) \in O(k) \cdot \left( \frac{1}{|\mathcal{L}^{m+1}|} \right)$.

# 4 From Sensitivity to Coreset

As previously stated, there are many frameworks (such as [14]) which provide a coreset given an upper bound of the sensitivity as in Lemma 3.3 and an upper bound for the VC-dimension as given in Section B.In what follows we present a general algorithm that uses such a framework.

**Overview of Algorithm 4** Given an $(n, m)$-set $\mathcal{L}$ of lines in $\mathbb{R}^d$, an integer $k \geq 1$, an error parameter $\varepsilon \in (0, 1)$, and a probability of failure $\delta \in (0, 1)$, Algorithm 4 computes an $\varepsilon$-coreset $(\mathcal{C}, v)$ for $\mathcal{L}$. First, the algorithm (Lines 1-7) recursively extracts a subset with known sensitivities using LS-DENSE until all sets are assigned a sensitivity, then calls an existing framework to compute the coreset (Line 8).

---

**Algorithm 4:** CORESET$(\mathcal{L}, k, \eta)$; see Theorem 4.1

**Input** : An $(n, m)$-set $\mathcal{L}$ of lines in $\mathbb{R}^d$, a positive integer $k \geq 1$, desired coreset size $\eta \geq 1$.
**Output** : A pair $(\mathcal{C}, v)$, where $\mathcal{C} \subseteq \mathcal{L}$ and $v : \mathcal{C} \to (0, \infty)$.

1   $\mathcal{L}^0 := \mathcal{L}^{m+1} := \mathcal{L}$
2   **while** $|\mathcal{L}^{m+1}| \geq 2$ **do**
3      $(\mathcal{L}^{m+1}, \mathcal{B}^m) := $ LS-DENSE$(\mathcal{L}^0, k)$// see Algorithm 3
4      **for** *every* $L \in \mathcal{L}^{m+1}$ **do**
5         $s(L) := \dfrac{1}{|\mathcal{L}^{m+1}|}$
6      $\mathcal{L}^0 := \mathcal{L}^0 \setminus \mathcal{L}^{m+1}$
7   **for** every set $L \in \mathcal{L}^0$ **do** $s(L) := 1$
8   $(\mathcal{C}, v) := $ CORESET-FW$(\mathcal{L}, s, \eta)$// see Algorithm 5
9   **Return**$(\mathcal{C}, v)$

---

**Theorem 4.1.** *Let $\mathcal{L}$ be an $(n, m)$-set of lines in $\mathbb{R}^d$, $k \geq 1$ be an integer, $\varepsilon, \delta \in (0, 1)$, and let $\eta \geq \left(\frac{m^{1.5} d \log n}{\varepsilon}\right)^2 (2k)^{cmk} + \log_2\left(\frac{1}{\delta}\right)$ be an integer, where $c$ is a sufficiently large constant that can be determined from the proof. Let $(\mathcal{C}, v)$ be the output of a call to* CORESET$(\mathcal{L}, k, \eta)$; *see Algorithm 4 in the Appendix. Then, Claims (i)–(ii) hold as follows.*

*(i) With probability at least $1 - \delta$, $(\mathcal{C}, v)$ is an $\varepsilon$-coreset of $\mathcal{L}$ for the $k$-line-sets of size $|\mathcal{C}| = \eta$.*
*(ii) The pair $(\mathcal{C}, v)$ can be computed in $n \log(n)(2k)^{O(mk)}$ time.*

By modifying Line 3 of Algorithm 4 to call CS-DENSE rather than LS-DENSE, as well as straightforward modifications over the input, one can achieve coreset construction for Colored-sets clustering as follows.

**Theorem 4.2.** *Let $k, m \geq 1$ be constant integers, let $\mathcal{P}$ be an $(n, m)$-ordered-set in $\mathbb{R}^d$, $\varepsilon, \delta \in (0, 1)$, and let $\eta \geq \left(\frac{m^{1.5} d \log n}{\varepsilon}\right)^2 (2k)^{cmk} + \log_2\left(\frac{1}{\delta}\right)$ be an integer, where $c$ is a sufficiently large constant that can be determined from the proof. There is an algorithm that given $\mathcal{P}, k, \varepsilon$ and $\delta$ returns with probability at least $1 - \delta$ an $\varepsilon$-coreset of $\mathcal{P}$ for colored-sets $k$-mean of size $\eta$ in $n \log(n)(2k)^{O(mk)}$ time.*

# 5 Experimental Results

We implemented our coreset construction algorithms for both Colored-sets and Line-sets. In this section we test there empirical performance on synthetic and real data. Open source is available in [32]. Since we provide the first coreset construction for the given problems, we compare our results to a single baseline - a random uniform sampling of the same number of points as in the coreset. Except for the case of $(n, 1)$-sets of lines where we also compare to the existing coreset for lines $k$-mean [35].

**Colored-sets data-sets**

**(i)** A *synthetic dataset* of size $n = 10000$ with outliers by sampling colored points from a set of Gaussians with different parameters corresponding to different colors. For each color there are two overlapping Gaussians for modeling the inliers of size $4950$ each and one Gaussian more than three standard deviations away of size $100$ for modeling the outliers.

**(ii)** The Reuters-21578 data-set from [30], which results in sets of points corresponding to each paragraph in each document. Additionally we divide paragraphs into categories: beginning (first third), main part (middle third) ending (last third). This results in each document being represented as a set of $m=3$ colored points corresponding to sets of paragraphs of different categories.

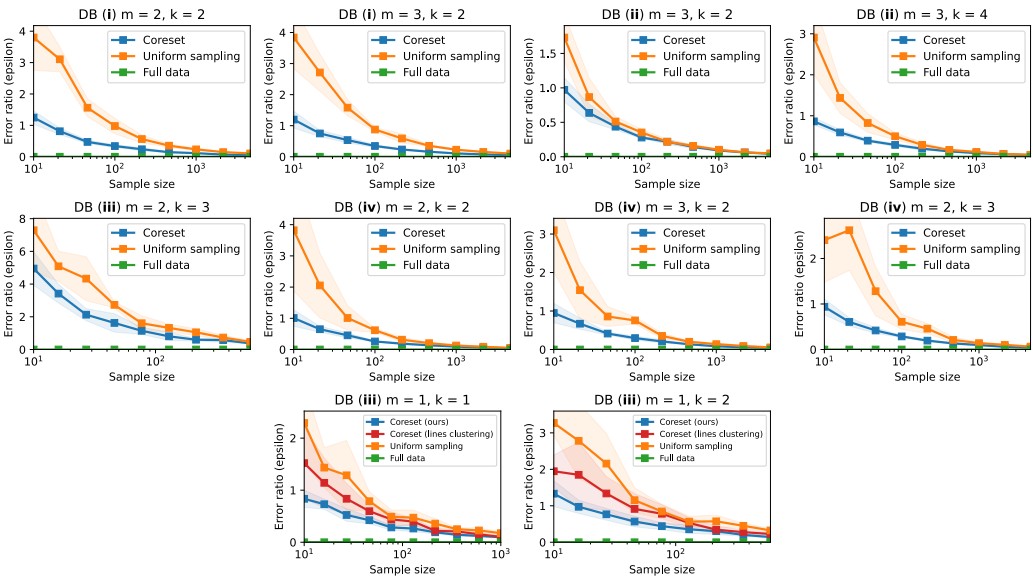

Figure 4: **Experimental results.** *See details in Section 5.*

**Line-sets data-sets**

**(iii)** A *synthetic dataset* that consists of the colored points dataset from the previous section, after removing the colors from the points. Each point was then turned into a line by assigning it to a random direction according to similar mixture of Gaussians.

**(iv)** We used California housing prices data-set [37] in witch we introduced uncertainty by removing two of the 9 dimensions each point. We removed one discrete value which created sets and one continuous value which created the lines.

**The experiment.** For each data set, and for different values of $m, k$, we conduct the following experiments. For every $\sigma \in \{10 \cdot 2^i\}_{i=0}^n$, a coreset $\mathcal{S}_1(\sigma)$ of size $\sigma$ using our algorithm, a uniform sample $\mathcal{S}_2(\sigma)$ of the data-set of size $\sigma$, and for the corresponding experiments a coreset $\mathcal{S}_3(\sigma)$ of size $\sigma$ using [35]. Then we generated a set $\mathcal{Q}, |Q| = 500$ queries: half of them are $k$-means that were computed using generalization of the EM heuristic, and the rest were randomly and uniformly sampled from the ground set. Finally, for every $i \in \{1, 2\}$ we computed the maximum approximation error

$$\mathcal{S}_i, \varepsilon_i(\sigma) = \max_{Q \in \mathcal{Q}} \frac{\left| \sum_{P \in \mathcal{P}} \tilde{D}(\mathcal{S}_i, Q) - \sum_{P \in \mathcal{P}} \tilde{D}(\mathcal{P}, Q) \right|}{\sum_{P \in \mathcal{P}} \tilde{D}(\mathcal{P}, Q)}$$

**Results** for the corresponding databases (i)–(iv) are shown in Fig. 4.

**Discussion** Our coresets out-preform uniform sampling in most of the experiments. As expected, when the data is uniformly distributed, the sensitivity of each point is close, and uniform sampling is already a coreset. Increasing $k$ and $m$ yields more isolated clusters, which explains why the error for the uniform samplings becomes higher compared to the coreset's error. As is in previous paper, the empirical error is very small compared to the theoretical worst-case bounds, even for small coresets.

## 6 Conclusion and Future Work

The paper suggests the first coresets for $k$ line-sets clustering via a reduction to coresets for colored-sets clustering of independent interest. We expect that this paper will open a line of research for many possible directions. For example, projectivedd clustering of sets. That is, replacing the *outputted* $k$ point centers by $k$ linear subspaces of $\mathbb{R}^d$, each of dimension $j \geq 1$. Another direction is to handle inputs sets of subspaces, each of dimension $j \geq 1$. Extending to non-linear shapes is another direction. Although, the main contribution of this paper is the theoretical breakthrough and results. However, we also expect that our code can be extended and applied for real-world systems that have similar problems, such as [24, 34].

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
