# A Appendix

# B Sensitivity Coresets

Every coreset in this paper is a weighted subset of its input, unlike other papers where the coreset may be a subset of a larger ground set. For example, points or lines in $\mathbb{R}^d$ instead of the input set. For clarity, the following definitions are a bit simpler than their cited versions due to this feature of our coresets. We also assume that the set of queries is the same for every input set and not a function of the resulting coreset. Such a generalization may be used to remove the dependency of the coreset on the dimension $d$; see Future Work in Section 6.

**Definition B.1** (Query space [9, 29]). *Let $P$ be a finite set. Let $Q$ be a (possibly infinite) set, called queries. Let $f : P \times Q \to [0, \infty)$ be a cost function. The tuple $(P, Q, f)$ is called a* query space.

In most papers, $P$ is a set of points in $\mathbb{R}^d$ or a metric space, and in some papers, $P$ is a set of lines. However, in this paper, $P$ is usually a set of sets, where each $p \in P$ corresponds to a set of lines or colored points. Every item in $Q$ is a set of $k$ centers, sometimes colored. Since this section is generic, we keep the classic notation defined above.

The following definition of VC-dimension is used in Theorem B.5 to bound the VC-dimension for our problems. Unfortunately, a different definition is used in the context of query spaces. Below we present the two definitions and show how they are equivalent.

**Definition B.2** (VC-dimension ). *Let $\mathcal{X}$ be a set, called* ground set, *and let $\mathcal{F}$ be a set of functions from $\mathcal{X}$ to $\{0, 1\}$. For a set $S \subseteq \mathcal{X}$, we call $S_f := \{x \in S | f(x) = 1\}$ the subset of $S$ induced by $f$. We say that $S$ is* shuttered *by $\mathcal{F}$ if and only if $|\{S_f | f \in \mathcal{F}\}| = 2^{|S|}$. The VC-dimension of $\mathcal{F}$ is the largest size of $S$ that is shuttered by $\mathcal{F}$.*

The definition of VC-dimension for query spaces is a straight forward generalization of the classic definition of VC-dimension in PAC-learning above.

**Definition B.3** (VC-dimension for query spaces [9]). *For a query space $(P, Q, f)$, a query $C \in Q$, and $r \in [0, \infty)$ we define*

$$\mathrm{range}(P, C, r) = \{p \in P | f(p, C) \leq r\}.$$

*Let $\mathrm{ranges}(P, Q, f) := \{\mathrm{range}(P, C, r) | C \in Q, r \geq 0\}$. The VC-dimension of the pair $(P, \mathrm{ranges}(P, Q, f))$ is the size $|S|$ of the largest subset $S \subseteq P$ such that*

$$\left|\left\{S \cap \mathrm{range}(P, C, r) | C \in Q, r \in [0, \infty)\right\}\right| = 2^{|S|}.$$

*The VC-dimension of the query space $(P, Q, f)$ is the VC-dimension of $(P, \mathrm{ranges}(P, Q, f))$.*

The following lemma show the relation between the pair of definitions above.

**Lemma B.4.** *Let $(P, Q, f)$ be a query space. We define*

$$\mathcal{H}((P, Q, f)) := \left\{ x \mapsto \begin{cases} 1, & f(x, C) \leq r \\ 0, & \text{otherwise} \end{cases} \middle| C \in Q, r \in [0, \infty) \right\}$$

*The VC-dimension of $\mathcal{H}((P, Q, f))$ is the VC-dimension of $(P, Q, f)$.*

*Proof.* Let $H = \mathcal{H}(P, Q, f)$ be as defined in Lemma B.4. Let $S \subseteq P$ be a set, and for every $h \in H$ let $S_h$ be the subset of $S$ induced by $h$. For every $C \in Q$ and $r \in [0, \infty)$, let $h'(x) = \begin{cases} 1, & f(x, C) \leq r \\ 0, & \text{otherwise} \end{cases}$ then every $x \in P$ is $x \in S_{h'} \Leftrightarrow x \in S \cap \mathrm{range}(P, C, r)$. Hence, for every $S' \subseteq S$ we have $S' \in \{S_h | h \in H\} \Leftrightarrow S' \in \{S \cap \mathrm{range}(P, C, r) | C \in Q, r \in [0, \infty)\}$. Finally, the VC-dimension of both, $H$ and $(P, Q, f)$ is the size of largest set $S$, such that $|\{S_h | h \in H\}| = |\{S \cap \mathrm{range}(P, C, r) | C \in Q, r \in [0, \infty)\}| = 2^{|S|}$. $\square$

**Lemma B.5** (Variant of Theorem 8.4, [3]). *Suppose $h$ is a function from $\mathbb{R}^d \times \mathbb{R}^n$ to $\{0, 1\}$ and let*

$$H = \{x \mapsto h(a, x) | a \in \mathbb{R}^d, x \in \mathbb{R}^n\}$$

*be the class determined by $h$. Suppose that $h$ can be computed by an algorithm that takes as an input a pair $(a, x) \in \mathbb{R}^d \times \mathbb{R}^n$ and returns $h(a, x)$ after no more than $t$ operations of the following types:*

- *the arithmetic operations $+, -, \times$, and $/$ on real numbers,*

- *jumps conditioned on $>, \geq, <, \leq, =$, and $\neq$ comparisons of real numbers, and*

- *outputs $0$ or $1$.*

*Then the $VC$-dimension of $H$ is $O\left(dt\right)$.*

We now bound the VC-dimension for our line-sets mean problem, whose query space is $\left(\mathcal{L}, \{C \subseteq \mathbb{R}^d | |C| = k\}, \tilde{D}\right)$.

**Lemma B.6.** *Let $\mathcal{L}$ be an $(n, m)$-set of lines in $\mathbb{R}^d$, $k \geq 1$ be an integer, and $Q = \{C \subseteq \mathbb{R}^d | |C| = k\}$. Let $(\mathcal{L}, Q, \tilde{D})$ be a line-sets clustering query space. Then the VC-dimension $d'$ of $(\mathcal{L}, Q, \tilde{D})$ is $d' \in O(md^2k^2)$.*

*Proof.* Let $\mathcal{Q} = \{C \subseteq \mathbb{R}^d | |C| = k\}$. Let $H = \mathcal{H}((\mathcal{L}, \mathcal{Q}, \tilde{D}))$ be as defined in Lemma B.4. Let $h : \mathcal{L} \times (Q \times \mathbb{R}) \to \{0, 1\}$ such that

$$h(L, (C, r)) = \begin{cases} 1, & if \ \tilde{D}(L, C) \geq r \\ 0, & Otherwise. \end{cases}$$

Note that, $H$ is determined by $h$. Assuming that each line $\ell \in L \in \mathcal{L}$ is represented by direction vector, it takes $t = O(mdk)$ arithmetic operations to evaluate $h$. Furthermore, any pair in $\mathcal{Q} \times \mathbb{R}$ can be represented as a vector in $(dk + 1)$-dimensional space. Hence by Lemma B.5, the $VC$-dimension of $H$ is $O(dk \cdot mdk) = O(md^2k^2)$. Finally, by Lemma B.4 the VC-dimension of $(\mathcal{L}, \mathcal{Q}, \tilde{D})$ is equal to the VC-dimension of $H$ and both are $O(md^2k^2)$. $\qquad\square$

In this paper we use the classic definition of sensitivity.

**Definition B.7** (Sensitivity). *Let $(P, Q, f)$ be a query space. The* sensitivity *function $s^* : P \to [0, \infty)$ of a query space $(P, Q, f)$ maps every $p \in P$ to*

$$s^*(p) := \sup_q \frac{f(p, q)}{\sum_{p' \in P} f(p', q)},$$

*where the supremum is over every $q \in Q$ such that $f(p, q) > 0$.*

*An upper bound for the sensitivity of such a query space is a function $s : P \to [0, \infty)$ such that $s(p) \geq s^*(p)$ for every $p \in P$.*

In the above definition we assumed that the supremum of an empty set is zero.

The following theorem proves that a coreset can be computed by sampling according to sensitivity of points. The size of the coreset depends on the total sensitivity and the complexity (VC-dimension) of the query space, as well as the desired error $\varepsilon$ and probability $\delta$ of failure.

**Theorem B.8** ([9]). *Let*

- *$(P, Q, f)$ be a query space, and $n = |P|$.*
- *$d'$ be the dimension of $(P, Q, f)$.*
- *$s : P \to [0, \infty)$ be a sensitivity bound of $(P, Q, f)$, and $t = \sum_{p \in P} s(p)$ be its total sensitivity.*
- *$\varepsilon, \delta \in (0, 1)$,*
- *$c > 0$ be a universal constant that can be determined from the proof,*
- *$\eta \geq \frac{c(t+1)}{\varepsilon^2} \left( d' \log(t+1) + \log \left( \frac{1}{\delta} \right) \right)$, and*
- *$(\mathcal{C}, v)$ be the output weighted set of a call to CORESET-FW$(P, s, \eta)$; see Algorithm 5.*

*Then $(i)$–$(ii)$ hold as follows.*

(i) *With probability at least $1 - \delta$, $(\mathcal{C}, v)$ is an $\varepsilon$-coreset of $(P, Q, f)$, whose size is $|\mathcal{C}| = \eta$; see Section 1.2.*

(ii) *$(\mathcal{C}, v)$ can be computed in $O(n)$ time, given $(P, s, \eta)$.*

---

**Algorithm 5:** CORESET-FW$(P, s, \eta)$

---

**Input** : A finite set $P \subseteq \mathbb{R}^d$, where $\sum_{p \in P} w(p) > 0$, a function $s : P \to [0, \infty)$, and an integer $\eta \geq 1$.
**Output** : A weighted set $(C, v)$.

**1** $s'(p) := s(p) + \frac{1}{|P|}$

**2** $\mathcal{C} := \left\{ p \in P \,\middle|\, \frac{s'(p)}{\sum_{q \in P} s'(q)} \geq \frac{1}{\eta} \right\}$ **for** every $p \in \mathcal{C}$ **do** $v'(p) := 1$

**3** $Q := P \setminus \mathcal{C}$

**4** $\eta' := \eta - |\mathcal{C}|$

**5** **for** $\eta'$ *iterations* **do**

**6** $\quad$ Sample a point $q$ from $Q$, where $q = p$ with probability $\Pr(p) := \dfrac{s'(p)}{\sum_{p' \in Q} s'(p')}$ $\mathcal{C} := \mathcal{C} \cup \{q\}$

**7** $\quad v'(q) := \dfrac{1}{\eta' \cdot \mathrm{pr}(q)}$

**8** **for** *every $p \in \mathcal{C}$* **do**

**9** $\quad v(p) := v'(p) \cdot \dfrac{|P|}{\sum_{q \in \mathcal{C}} v'(q)}$

**10** **Return** $(C, v)$

---

**Theorem B.9** (Restatement of Theorem 4.1). *Let $\mathcal{L}$ be an $(n, m)$-set of lines in $\mathbb{R}^d$, $k \geq 1$ be an integer, $\varepsilon, \delta \in (0, 1)$, and let*

$$\eta \geq \left( \frac{m^{1.5} d \log n}{\varepsilon} \right)^2 (2k)^{cmk} + \log \left( \frac{1}{\delta} \right)$$

*be an integer, where $c$ is sufficiently large constant that can be determined from the proof. Let $(\mathcal{C}, v)$ be the output of a call to CORESET$(\mathcal{L}, k, \eta)$; see Algorithm 4. Then, Claims $(i)$–$(ii)$ hold as follows.*

(i) *With probability at least $1 - \delta$, $(\mathcal{C}, v)$ is an $\varepsilon$-coreset of $\mathcal{L}$ for the $k$-line-sets of size $|\mathcal{C}| = \eta$; see Section 1.2.*

(ii) *The pair $(\mathcal{C}, v)$ can be computed in $n \log(n)(2k)^{O(mk)}$ time.*

*Proof.* Let $J$ denote the number of "while" iterations in Line 2 of Algorithm 4. For every $j \in [J]$, let $\mathcal{L}_j^0$, $\mathcal{L}_j^{m+1}$ and $\mathcal{B}_j^m$ denote, respectively, the sets $\mathcal{L}^0$, $\mathcal{L}^{m+1}$ and $\mathcal{B}^m$ during the execution of Line 3 at the $j$th "while" iteration.

Let $j \in [J]$. The pair $(\mathcal{L}_j^{m+1}, \mathcal{B}_j^m)$ is the output of a call to LS-DENSE$(\mathcal{L}_j^0, k)$. Hence, by Lemma 3.4, with an appropriate choice of $b \in O(k)$ (determined from the proof of Lemma 3.4), for every $L \in \mathcal{L}_j^{m+1}$ its value $s(L)$ that is defined in Lines 5 satisfies, for every $C \subseteq \mathbb{R}^d$ of size $|C| = k$, such

that $\tilde{D}(L, C) > 0$,

$$b \cdot s(L) = \frac{b}{\left|\mathcal{L}_j^{m+1}\right|} \geq \frac{\tilde{D}(L, C)}{\sum_{Q \in \mathcal{L}_j^0} \tilde{D}(Q, C)} \geq \frac{\tilde{D}(L, C)}{\sum_{Q \in \mathcal{L}} \tilde{D}(Q, C)}, \tag{5}$$

where the first inequality is by Lemma 3.4, and the second inequality holds since $\mathcal{L}_j^0 \subseteq \mathcal{L}$. Since (5) holds for every $C$, and $L \in \mathcal{L}$ we have

$$b \cdot s(L) \geq \sup_{\substack{C \subseteq \mathbb{R}^d, |C| = k \\ \tilde{D}(L, C) > 0}} \frac{\tilde{D}(L, C)}{\sum_{Q \in \mathcal{L}} \tilde{D}(Q, C)} \geq S_{\mathcal{L}, k}(L). \tag{6}$$

By Line 8 of Algorithm 4, the pair $(\mathcal{C}, v)$ is the output of Algorithm 5. By (6) $b \cdot s(L)$ is an upper bound to the sensitivity of $\mathcal{L}$. Note that in Line 8 we call Algorithm 5 with $s$ and not with $b \cdot s$, however, the distribution is the same due to the scaling in Line 6 of Algorithm 5. By Theorem B.8 $(\mathcal{C}, v)$, is an $\varepsilon$-coreset with probability at least $1 - \delta$ if

$$\eta \geq \frac{c(t+1)}{\varepsilon^2} \left( d' \log(t+1) + \log\left(\frac{1}{\delta}\right) \right), \tag{7}$$

where $c$ is a constant that can be determined from the proof, $d'$ is the VC-dimension of the query space $\left( \mathcal{L}, \{X \subseteq \mathbb{R}^d \mid |X| = k\}, \tilde{D} \right)$, and $t = \sum_{L \in \mathcal{L}} s(L)$ is the total sensitivity of $\mathcal{L}$.

In order to bound the total sensitivity, given a set $\mathcal{L}'$ we first determine the size of fraction of sets from $\mathcal{L}'$ that outputted by the execution of LS-DENSE$(\mathcal{L}', k)$; see Algorithm 3. Let $\tau = \frac{1}{20}$, and $a = \frac{4}{1-\tau}$. By Line 5 of Algorithm 3, for every $i \in [m]$

$$\left| \mathcal{L}^i \right| = \left\lceil \frac{\left| \mathcal{L}^{i-1} \right|}{ak} \right\rceil. \tag{8}$$

Therefore, by induction

$$\left| \mathcal{L}^m \right| \geq \frac{\left| \mathcal{L}^0 \right|}{(ak)^m}. \tag{9}$$

The argmin set in Line 10 is the set of lines that is retuned in the fist iteration of Algorithm 2. Similarly to (9), by induction over Line 6 of Algorithm 1 we have

$$\left| \mathcal{L}^{m+1} \right| \geq \frac{\left| \mathcal{L}^m \right|}{(ak)^{mk}}, \tag{10}$$

and by combining (9) with (10) we get

$$\left| \mathcal{L}^{m+1} \right| \geq \frac{\left| \mathcal{L}^0 \right|}{(ak)^{mk+m}}. \tag{11}$$

Combining (11) and Line 3 of Algorithm 4 yields

$$\left| \mathcal{L}_j^{m+1} \right| \geq \frac{\left| \mathcal{L}_j^0 \right|}{(ak)^{mk+m}}. \tag{12}$$

By (12) and Line 6 of Algorithm 4

$$\begin{aligned}
\left| \mathcal{L}_{j+1}^0 \right| \leq \left| \mathcal{L}_j^0 \right| - \left| \mathcal{L}_j^{m+1} \right| &\leq \left| \mathcal{L}_j^0 \right| - \frac{\left| \mathcal{L}_j^0 \right|}{(ak)^{mk+m}} \\
&= \left| \mathcal{L}_j^0 \right| \left( 1 - \frac{1}{(ak)^{mk+m}} \right) \\
&= \left| \mathcal{L}_1^0 \right| \left( 1 - \frac{1}{(ak)^{mk+m}} \right)^j \\
&= n \left( 1 - \frac{1}{(ak)^{mk+m}} \right)^j,
\end{aligned} \tag{13}$$

where the second inequality is by (12). Combining the fact that $\left|\mathcal{L}_j^0\right| \geq 1$ with (13), we conclude that

$$J \leq (ak)^{mk+m} \log_2 n. \tag{14}$$

Therefore, the total sensitivity $t$ is bounded by

$$
\begin{aligned}
t &\leq \sum_{L \in \mathcal{L}} s(L) = \sum_{j=1}^{J} \left( \sum_{L \in \mathcal{L}_j^{m+1}} \frac{b}{\left|\mathcal{L}_j^{m+1}\right|} \right) + c = \sum_{j \in [J]} \left( \left|\mathcal{L}_j^{m+1}\right| \cdot \frac{b}{\left|\mathcal{L}_j^{m+1}\right|} \right) + c \\
&= \sum_{j \in [J]} b + c = Jb + c \leq (ak)^{mk+m+1} \log_2 n,
\end{aligned}
$$

where $c \geq 1$ is a constant. By combining with (7), we get that the pair $(\mathcal{C}, v)$ is an $\varepsilon$-coresets for $(\mathcal{L}, k)$ if

$$\eta \geq \frac{(ak)^{mk+m+1} \log_2 n}{\varepsilon^2} \left( \log_2 \left( (ak)^{mk+m+1} \log n \right) d' + \log_2 \left( \frac{1}{\delta} \right) \right).$$

where $d' \in O(md^2 k^2)$ (by Lemma B.6) is the VC-dimension of the sets clustering query space of the lines set clustering problem.

**Running time:** Consider a call CS-DENSE$(\mathcal{P}, k)$ to Algorithm 1 where $\mathcal{P}$ is an $(n, m)$-set. For every $i \in [k], j \in [m]$ the $i, j$th iteration of the "for" loops takes $O\left( dn \left( \frac{1}{4k} \right)^{jm+i-1} + dk^4 \right)$ time. Summing over all the $mk$ iterations yields a total running time of $O\left( dn + dmk^5 \right)$.

Consider a call LS-DENSE$(\mathcal{L}, k)$ to Algorithm 3, where $\mathcal{L}$ is an $(n, m)$-set of lines in $\mathbb{R}^d$. For every $i \in [m]$, the $i$th iteration of the "for" loop at Line 3 takes $O\left( dn \left( \frac{1}{4k} \right)^{j-1} + dk^4 \right)$ time. Summing over all the $m$ iterations yields a total running time of $O(dn + dk^4)$. Combined with the call to CS-DENSE inside the call to GROUPED-SENSITIVITY at Line 9 the overall time is $O(dn + dmk^5)$.

There are $J$ calls to LS-DENSE$(\mathcal{L}^0, k)$ at Line 3 of Algorithm 4, which dominates the running time of this algorithm (in each of the $J$ iterations of the "while" loop). The set $\mathcal{L}^0$ at the $j$th call is of size $\left|\mathcal{L}_j^0\right| \in O\left( n \left( 1 - \frac{1}{4k} \right)^{i-1} \right)$. Therefore, the $j$th call takes $O\left( d \left|\mathcal{L}_j^0\right| + dmk^5 \right)$ time. Summing this running time over every $j \in [J]$, where $J \leq (ak)^{mk+m} \log n$ by (14), yields a total running time of

$$J \cdot dmk^5 + dn \sum_{i=1}^{J} \left( 1 - \frac{1}{4k} \right)^{i-1} \in dn \log_2(n)(ak)^{o(mk)},$$

as claimed in $(ii)$. $\qquad\square$

**Theorem B.10** (Restatement of Thorem 4.2). *Let $\mathcal{P}$ be an $(n, m)$-ordered-set in $\mathcal{X}$, let $k \geq 1$ be an integer, $\varepsilon, \delta \in (0, 1)$, and let*

$$\eta \geq \left( \frac{m^{1.5} d \log n}{\varepsilon} \right)^2 (2k)^{cmk} + \log \left( \frac{1}{\delta} \right)$$

*be an integer, where $c$ is sufficiently large constant that can be determined from the proof. There is an algorithm that given $\mathcal{P}, k, \varepsilon$ and $\delta$ return with probability at least $1 - \delta$ an $\varepsilon$-coreset of $\mathcal{P}$ for colored-sets $k$-mean of size $\eta$ in $n \log(n)(2k)^{O(mk)}$ time.*

Such algorithm is achieved by little variation over Algorithm 4 and using CS-DENSE instead LS-DENSE.

*Proof.* Similar and can be deduced from the proof of Theorem B.9. $\qquad\square$

## C Algorithms Correctness

### C.1 Correctness of Algorithm 1

The goal of Algorithm 1 is to find a small family (set) $\mathcal{P}'$ of $\Theta(n)$ sets that are sufficiently mutually close that they can be replaced with multiple copies of the same set with little to no effect on the cost.

In that case, all of their sensitivities would be $\sim \frac{1}{|\mathcal{P}'|}$. The proof of this lemma is by case analesis of two cases. The first case assumes that one of the centers in the query is close to $\mathcal{P}'$. In this case, the sets in $\mathcal{P}'$ are not affecting the cost at all. In the other case, we assume that all of the centers are far from $\mathcal{P}'$ and then we try to show that they are much closer to each other than to the centers relative to the cost of the centers.

*Proof of Lemma 3.3.* Let $C_w \subseteq \mathcal{X} \times [0, \infty) \times [m]$ be a set of $|C_w| = k$ weighted colored centers. Consider the variables $\tau, \mathcal{P}_1^m, \ldots, \mathcal{P}_k^m$ and $b_1^1, \ldots, b_k^m$ that are computed during the execution of CS-DENSE$(\mathcal{P}, k)$. For every $i \in [m]$ and $r \in [k]$, identify $b_r^i = (x_r^i, j)$. Without the loss of generality, assume that $j = 1$. Therefore, for every $P \in \mathcal{P}$, $i \in [m]$ and $r \in [k]$, we have

$$\tilde{D}(\overline{\text{proj}}(P, (b_r^1, \ldots, b_r^{i-1})), b_r^i) = \tilde{D}\left(P, \left(x_r^i, i\right)\right).$$

For the rest of the proof, let $\bar{b}_r^i = (x_r^i, i)$.

Let $\mathcal{P}_0^0 = \ldots, \mathcal{P}_0^m := \mathcal{P}$. We say that an ordered-set $P \in \mathcal{P}$ is *served* by a colored weighted center $(c, w, t) \in C_w$ if $\tilde{D}(P, C_w) = \tilde{D}(P, (c, w, t))$. For every $i \in [k+1]$, let $(c_i, w_i, t_i) \in C$ denote a center that serves at least $|\mathcal{P}_{i-1}^m|/k$ sets from $\mathcal{P}_{i-1}^m$, and let $\mathcal{P}_i$ denote the sets of $\mathcal{P}$ that are served by $(c_i, w_i, t_i)$. For every $r \in [k]$ and $\ell \in [m]$, let

$$\mathcal{Q}_{r,\ell} \in \underset{\mathcal{Q} \subseteq \mathcal{P}_r^{\ell-1}, |\mathcal{Q}| = \frac{(1-\tau) \cdot \left|\mathcal{P}_r^{\ell-1}\right|}{k}}{\arg \min} \sum_{Q \in \mathcal{Q}} \tilde{D}(Q, \bar{b}_r^\ell), \tag{15}$$

and denote $\tilde{D}_{r,\ell}^* = \sum_{Q \in \mathcal{Q}_{r,\ell}} \tilde{D}(Q, \bar{b}_r^\ell)$.

Since for every $i \in [k]$ we have $|\mathcal{P}_i \cap \mathcal{P}_{i-1}^m| \geq |\mathcal{P}_{i-1}^m|/k$, by the definition of the robust median

$$\sum_{Q \in \mathcal{P}_i \cap \mathcal{P}_{i-1}^m} \tilde{D}(Q, c_i) \geq \tilde{D}^*\left(\mathcal{P}_{i-1}^m, \frac{1}{k}\right). \tag{16}$$

Let $P = (p_1, \ldots, p_m) \in \mathcal{P}$, such that $\tilde{D}(P, C) > 0$. The rest of the proof uses the following case analysis, *(i)* there is an index $i \in [k]$ such that

$$\tilde{D}(P, c_i) \leq \frac{16\phi\rho\alpha\tilde{D}_{i,t_i}^*}{|\mathcal{Q}_{k,m}|}, \tag{17}$$

where $\phi$ and $\rho$ are constants defined in Lemma 2.2 , and *(ii)* Otherwise.

**Proof for Case *(i)*:** By (17),

$$\frac{\tilde{D}(P, C)}{\sum_{Q \in \mathcal{P}} \tilde{D}(Q, C)} \leq \frac{w_i \tilde{D}(p_{t_i}, c_i)}{\sum_{Q \in \mathcal{P}_i} \tilde{D}(Q, C)} \tag{18}$$

$$= \frac{\tilde{D}(p_{t_i}, c_i)}{\sum_{(q_1, \ldots, q_m) \in \mathcal{P}_i} \tilde{D}(q_{t_i}, c_i)} \tag{19}$$

$$\leq \frac{16\phi\rho\alpha\tilde{D}_{i,t_i}^*/|\mathcal{Q}_{k,m}|}{\sum_{(q_1, \ldots, q_m) \in \mathcal{P}_i \cap \mathcal{P}_{i-1}^m} \tilde{D}(q_{t_i}, c_i)} \tag{20}$$

$$\leq \frac{16\phi\rho\alpha\tilde{D}_{i,t_i}^*/|\mathcal{Q}_{k,m}|}{\tilde{D}_{i,t_i}^*/\alpha} \tag{21}$$

$$\leq \frac{16\phi\rho\alpha^2}{|\mathcal{P}_k^m|}, \tag{22}$$

where (18) holds since $\mathcal{P}_i$ is a subset of $\mathcal{P}$, (20) holds by the assumption of Case (i), (21) holds by combining (16) and Definition 2.5 $(i)$, and (22) holds since $|\mathcal{Q}_{k,m}| \subseteq |\mathcal{P}_k^m|$ .

**Proof for Case *(ii)*:** By the pigeonhole principle, $c_i = c_j$ for some $i, j \in [k+1]$, such that $i < j$. Put $Q = (q_1, \ldots, q_m) \in \mathcal{P}_j \cap \mathcal{P}_{j-1}^m$ and note that $P \in \mathcal{P}_k^m \subseteq \mathcal{P}_{j-1}^m$. By Markov's inequality, for every $\ell \in [m]$ we have

$$\tilde{D}(P, \bar{b}_{j-1}^\ell), \tilde{D}(Q, \bar{b}_{j-1}^\ell) \leq \frac{2\tilde{D}_{j-1,\ell}^*}{|\mathcal{Q}_{j-1,\ell}|}. \tag{23}$$

By (23) and Lemma 2.2 $(i)$, for every $\ell \in [m]$

$$\tilde{D}(P,Q) \le \rho \left( \tilde{D}(p_\ell, \bar{b}_{j-1}^\ell) + \tilde{D}(q_\ell, \bar{b}_{j-1}^\ell) \right) \le \frac{4\rho \tilde{D}_{j-1,\ell}^*}{|\mathcal{Q}_{j-1,\ell}|}. \tag{24}$$

Combining the last inequality with Definition 2.1$(ii)$, yields

$$\tilde{D}(P, (c_j, t_j)) - \tilde{D}(Q, (c_j, t_j)) = \tilde{D}(p_{t_i}, c_j) - \tilde{D}(q_{t_i}, c_j)$$

$$\le \phi \tilde{D}(p_{t_i}, q_{t_i}) + \frac{\tilde{D}(p_{t_i}, c_j)}{4} \tag{25}$$

$$\le \frac{4\phi\rho \tilde{D}_{j-1,t_j}^*}{|\mathcal{Q}_{j-1,t_j}|} + \frac{\tilde{D}(p_{t_i}, c_j)}{4} \tag{26}$$

$$\le \frac{4\phi\rho \tilde{D}_{j-1,t_j}^*}{|\mathcal{Q}_{k,m}|} + \frac{\tilde{D}(p_{t_i}, c_j)}{4}, \tag{27}$$

where (25) is followed from Lemma 2.2$(ii)$ and $(i)$ respectively. Finally (26) is obtained after plugging (24) in (25), and (27) is since $|\mathcal{Q}_{k,m}| \le |\mathcal{Q}_{i,j}|$ for every $i \in [k]$ and $j \in [m]$.

By the assumption of Case$(ii)$, for every $r \in [m]$,

$$\tilde{D}(P, c_j) = \tilde{D}(P, c_i) > \frac{16\phi\rho\alpha \tilde{D}_{j,r}^*}{|\mathcal{Q}_{k,m}|}.$$

Hence

$$\frac{\tilde{D}(P, c_j)}{4} > \frac{4\phi\rho\alpha \tilde{D}_{j,r}^*}{|\mathcal{Q}_{k,m}|}.$$

Combining with (27) yields

$$\tilde{D}(P, c_j) - \tilde{D}(Q, c_j) \le \frac{\tilde{D}(P, c_j)}{4} + \frac{\tilde{D}(P, c_j)}{4} = \frac{\tilde{D}(P, c_j)}{2}, \tag{28}$$

that is $\tilde{D}(Q, c_j) \ge \tilde{D}(P, c_j)/2$. Hence,

$$\frac{\tilde{D}(P, C)}{\sum_{P' \in \mathcal{P}} \tilde{D}(P', C)} \le \frac{\tilde{D}(P, c_j)}{\sum_{Q \in \mathcal{P}_j \cap \mathcal{P}_{j-1}^m} \tilde{D}(Q, c_j)}$$

$$\le \frac{2\tilde{D}(P, c_j)}{\tilde{D}(P, c_j)|\mathcal{P}_j \cap \mathcal{P}_{j-1}^m|}$$

$$\le \frac{2k}{|\mathcal{P}_{j-1}^m|} \tag{29}$$

$$\le \frac{2k}{|\mathcal{P}_k^m|},$$

where (29) holds since $\mathcal{P}_j$ serves at least $\left|\mathcal{P}_{j-1}^m\right|/k$ sets of $\mathcal{P}_{j-1}^m$. $\qquad \square$

**Corollary C.1.** *Let $\mathcal{P}$, $k$ and $(\mathcal{P}_k^m, \mathcal{B}_k^m)$ be as in Lemma 3.3, and let* lip, $r$ *be as defined in Definition 2.1. Then, for every set $P \in \mathcal{P}_k^m$ and a set $C_w \subseteq \mathcal{X} \times [0, \infty) \times [m]$ of $|C| = k$ weighted colored centers such that $\tilde{D}(P, C) > 0$, we have*

$$\frac{\mathrm{lip}(\tilde{D}(P, C))}{\sum_{P' \in \mathcal{P}} \mathrm{lip}(\tilde{D}(P', C))} \le \frac{2^r k}{|\mathcal{P}_k^m|}. \tag{30}$$

*Proof.* In what follows, we use the variables and notations from proof of Lemma 3.3. The proof is is similar to the proof of Lemma 3.3, and is via the following case analysis. *(i)* there is an index $i \in [k]$ such that:

$$\tilde{D}(P, c_i) \le \frac{16\phi\rho\alpha \tilde{D}_{i,t_i}^*}{|\mathcal{Q}_{k,m}|}, \tag{31}$$

and *(ii)* Otherwise.

**Case *(i)*:** By (31),

$$\frac{\text{lip}(\tilde{D}(P,C))}{\sum_{Q\in\mathcal{P}}\text{lip}(\tilde{D}(Q,C))} \leq \frac{16\phi\rho\alpha^2}{|\mathcal{P}_k^m|},\tag{32}$$

similar to (22).

**Variation over Case *(ii)*:** By (28) we have,

$$2\tilde{D}(Q,c_j) \geq \tilde{D}(P,c_j),$$

and by Definition 2.1,

$$2^r\text{lip}\left(\tilde{D}(Q,c_j)\right) \geq \text{lip}\left(\tilde{D}(P,c_j)\right).\tag{33}$$

Hence,

$$\begin{aligned}
\frac{\text{lip}\left(\tilde{D}(P,C)\right)}{\sum_{P'\in\mathcal{P}}\text{lip}\left(\tilde{D}(P',C)\right)} &\leq \frac{\text{lip}\left(\tilde{D}(P,C)\right)}{\sum_{Q\in\mathcal{P}_j\cap\mathcal{P}_{j-1}^m}\text{lip}\left(\tilde{D}(Q,C)\right)}\\
&\leq \frac{w_i^r\text{lip}\left(\tilde{D}(P,C)\right)}{w_i^r\sum_{Q\in\mathcal{P}_j\cap\mathcal{P}_{j-1}^m}\text{lip}\left(\tilde{D}(Q,C)\right)}\\
&\leq \frac{2^r\text{lip}\left(\tilde{D}(P,c_j)\right)}{\text{lip}\left(\tilde{D}(P,c_j)\right)|\mathcal{P}_j\cap\mathcal{P}_{j-1}^m|}\\
&\leq \frac{2^r k}{|\mathcal{P}_{j-1}^m|}\\
&\leq \frac{2^r k}{|\mathcal{P}_k^m|},
\end{aligned}\tag{34}$$

where (34) holds since $\mathcal{P}_j$ serves at least $\left|\mathcal{P}_{j-1}^m\right|/k$ sets of $\mathcal{P}_{j-1}^m$. $\qquad\square$

The following claim has nothing to do with the correctness of Algorithm 1. However, it will be used later in the proof of Lemma 3.4.

**Claim C.2.** *Let $\mathcal{P}$ be an $(n,m)$-ordered-set in $(\mathcal{X},D)$, and $k\geq 1$ be an integer. For every integer $i\in[m]$, and ordered set $P\in\mathcal{P}$, we have*

$$\sup_{\substack{C_w\subseteq\mathcal{X}\times\mathbb{R}\times[m]\\|C_w|=k,\ \tilde{D}(P,C_w)>0}}\frac{\tilde{D}(P,C_w)}{\sum_{Q\in\mathcal{P}}\tilde{D}(Q,C_w)} \geq \sup_{\substack{C_w\subseteq\mathcal{X}\times\mathbb{R}\times[i]\\|C_w|=k,\ \tilde{D}(P,C_w)>0}}\frac{\tilde{D}((p_1,\ldots,p_i),C_w)}{\sum_{(q_1,\ldots,q_m)\in\mathcal{P}}\tilde{D}((q_1,\ldots,q_i),C_w)},\tag{35}$$

*i.e., the sensitivity of any prefix with respect to all the other sets prefixes is smaller than the sensitivity of the original set with respect to the original family.*

*Proof.* Let $i\in[m]$ be an integer. Let $C\subseteq\mathcal{X}\times(0,\infty)\times[i]$ be a set that maximizes the right hand side of (35), i.e.,

$$\sup_{\substack{C_w\subseteq\mathcal{X}\times\mathbb{R}\times[i]\\|C_w|=k,\ \tilde{D}(P,C_w)>0}}\frac{\tilde{D}((p_1,\ldots,p_i),C_w)}{\sum_{(q_1,\ldots,q_m)\in\mathcal{P}}\tilde{D}((q_1,\ldots,q_i),C_w)} = \frac{\tilde{D}((p_1,\ldots,p_i),C)}{\sum_{(q_1,\ldots,q_m)\in\mathcal{P}}\tilde{D}((q_1,\ldots,q_i),C)}.$$

Then

$$\sup_{\substack{C_w\subseteq\mathcal{X}\times\mathbb{R}\times[m]\\|C_w|=k,\ \tilde{D}(P,C_w)>0}}\frac{\tilde{D}(P,C_w)}{\sum_{Q\in\mathcal{P}}\tilde{D}(Q,C_w)} \geq \frac{\tilde{D}(P,C)}{\sum_{Q\in\mathcal{P}}\tilde{D}(Q,C)} =\tag{36}$$

$$\sup_{\substack{C_w\subseteq\mathcal{X}\times\mathbb{R}\times[i]\\|C_w|=k,\ \tilde{D}(P,C_w)>0}}\frac{\tilde{D}((p_1,\ldots,p_i),C_w)}{\sum_{(q_1,\ldots,q_m)\in\mathcal{P}}\tilde{D}((q_1,\ldots,q_i),C_w)}\tag{37}$$

where (36) holds since $\mathcal{X}\times(0,\infty)\times[i]\subseteq\mathcal{X}\times(0,\infty)\times[m]$, and (37) is by the definition of $C$. $\quad\square$

## C.2 Correctness of Algorithm 2

**Claim C.3.** *Let $p$ be a point on a line $\ell$ in $\mathbb{R}^d$, and let $\mathcal{S} = \{c \in \mathbb{R}^d \mid \|p - c\| = 1\}$ denote the unit sphere that is centered at $p$; see Fig. 5. Then, for every point $q \in \mathcal{S}$, we have*

$$\text{dist}(\ell, q) \leq \sqrt{2} \cdot \text{dist}(\mathcal{S} \cap \ell, q).$$

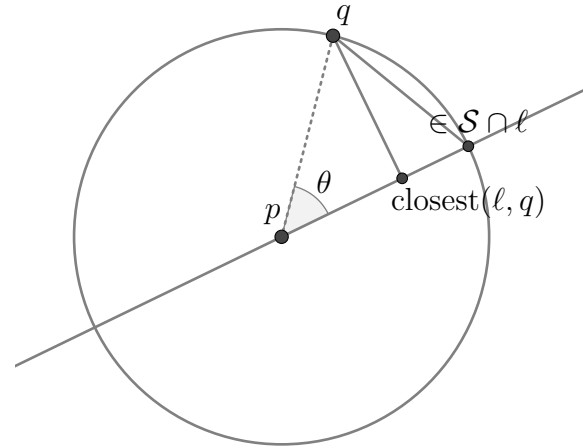

Figure 5

*Proof of Claim C.3.* Let $a, b$ be a pair of unit vectors in $\mathbb{R}^d$ such that $a^T b \geq 0$.

$$\|a - b\| = \sqrt{\|a - b\|^2} = \sqrt{2 \|1 - a^T b\|} \tag{38}$$

$$\leq \sqrt{2 \|1 - (a^T b)^2\|} = \sqrt{2 \text{dist}^2(a, \text{sp}(b))} = \sqrt{2} \text{dist}(a, \text{sp}(b)) \tag{39}$$

Then if $p$ were on the origin by substitute $a = q$ and $b = \mathcal{S} \cap \ell$ we get

$$\text{dist}(\ell, q) \leq \sqrt{2} \text{dist}(\mathcal{S} \cap \ell, q).$$

$\square$

**Lemma C.4.** *Let $\mathcal{L}$ be an $(n, m)$-set of lines, and $B = (b_1, \ldots, b_m)$ be a set of $m$ points, both in $\mathbb{R}^d$, such that for every set $L = (\ell_1, \ldots, \ell_m) \in \mathcal{L}$ and every $i \in [m]$, the line $\ell_i$ intersects $b_i$. Let $k \geq 1$ be an integer, and let $s : \mathcal{L} \to (0, 1]$ be the output of call to* GROUPED-SENSITIVITY$(\mathcal{L}, B, k)$;*see Algorithm 2. Then, for every $L \in \mathcal{L}$, we have*

$$s(L) \geq S_{\mathcal{L}, k}(L).$$

*Proof of Lemma C.4.* Define $P(L)$ for every $L \in \mathcal{L}$, as in Algorithm 2. For every line $\ell \subseteq \mathbb{R}^d$, $p' \in \ell$, and $p \in \mathbb{R}^d \setminus \{p'\}$, by Thales Theorem, we have

$$\text{dist}(\ell, p) = \text{dist}(p, p') \cdot \text{dist}\left(\ell, p' + \frac{p - p'}{\text{dist}(p, p')}\right). \tag{40}$$

Let $L = (\ell_1, \ldots, \ell_m) \in \mathcal{L}$, and $C \subseteq \mathbb{R}^d$ be a set of $|C| = k$ centers such that $\tilde{D}(L, C) > 0$, and recall that, by (4), for every set $P$ in $\mathcal{X}$ let $\text{closest}(C, P)$ denote the only point in $\text{closest}(C, P, \frac{1}{|C|})$ consists of the closest point in $C$ to a set in $P$. Hence,

$$\begin{aligned}
\tilde{D}(L, C) &= \min_{i \in [m]} \tilde{D}(\ell_i, C) \\
&= \min_{i \in [m]} \text{lip}\left(\text{dist}(\ell_i, \text{closest}(C, \ell_i))\right) \\
&= \min_{i \in [m]} \text{lip}\left(\text{dist}(\text{closest}(C, \ell_i), b_i) \cdot \text{dist}\left(\ell_i, b_i + \frac{\text{closest}(C, \ell_i) - b_i}{\text{dist}(\text{closest}(C, \ell_i), b_i)}\right)\right), \tag{41}
\end{aligned}$$

where (41) holds by (40).

For every $i \in [m]$, and every point $q \in \mathbb{R}^d \setminus \{b_i\}$, the point $b_i + \frac{q - b_i}{\|q - b_i\|}$ is in $\mathbb{S}_i$, i.e., on the unit sphere that is centered at $b_i$. By Claim C.3,

$$
\frac{\tilde{D}(L, C)}{\sum_{L' \in \mathcal{L}} \tilde{D}(L', C)} \leq
$$

$$
\frac{\sqrt{2^r} \cdot \min_{i \in [m]} \text{lip}\left(\text{dist}(\text{closest}(C, \ell_i), b_i) \cdot \text{dist}\left(\ell_i \cap \mathbb{S}_i, b_i + \frac{\text{closest}(C, \ell_i) - b_i}{\|\text{closest}(C, \ell_i) - b_i\|}\right)\right)}{\sum_{(\ell_1', \ldots, \ell_m') \in \mathcal{L}} \min_{j \in [m]} \text{lip}\left(\text{dist}(\text{closest}(C, \ell_j'), b_j) \cdot \text{dist}\left(\ell_j' \cap \mathbb{S}_j, b_j + \frac{\text{closest}(C, \ell_j') - b_j}{\|\text{closest}(C, \ell_j') - b_j\|}\right)\right)}. \tag{42}
$$

Let

$$
C' = \left\{ \left( b_i + \frac{c - b_i}{\|c - b_i\|}, \|c - b\|, i \right) \,\middle|\, i \in [m], c \in C \right\}. \tag{43}
$$

Hence, we can reformulate the right hand side of (42) to

$$
\frac{\tilde{D}(L, C)}{\sum_{L' \in \mathcal{L}} \tilde{D}(L', C)} \leq \frac{\sqrt{2^r} \cdot \min_{(c, w, t) \in C'} \text{lip}\left(w \cdot \text{dist}\left(\ell_t \cap \mathbb{S}_t, c\right)\right)}{\sum_{(\ell_1', \ldots, \ell_m') \in \mathcal{L}} \min_{(c', w', t') \in C'} \text{lip}\left(w' \cdot \text{dist}\left(\ell_{t'} \cap \mathbb{S}_{t'}, c'\right)\right)}. \tag{44}
$$

Since $C' \subseteq \mathbb{R}^d \times (0, \infty) \times [m]$, and the cardinality of the set $C'$ is at most $|C| \cdot |B| = mk$, we have

$$
S_{\mathcal{L}, k}(L) = \sup_{\substack{C \subseteq \mathbb{R}^d, |C| = k \\ \tilde{D}(L, C) > 0}} \frac{\tilde{D}(L, C)}{\sum_{L' \in \mathcal{L}} \tilde{D}(L', C)} \tag{45}
$$

$$
\leq \sup_{\substack{C_w \subseteq \mathbb{R}^d \times (0, \infty) \times [m] \\ |C_w| = mk}} \frac{\sqrt{2^r} \cdot \min_{(c, w, t) \in C_w} \text{lip}\left(w \cdot \text{dist}\left(\ell_t \cap \mathbb{S}_t, c\right)\right)}{\sum_{\{\ell_1', \ldots, \ell_m'\} \in \mathcal{L}} \min_{(c', w', t') \in C_w} \text{lip}\left(w' \cdot \text{dist}\left(\ell_{t'} \cap \mathbb{S}_{t'}, c'\right)\right)} \tag{46}
$$

$$
\geq \sup_{\substack{C_w \subseteq \mathbb{R}^d \times (0, \infty) \times [2m] \\ |C_w| = 2mk}} \frac{\sqrt{2^r} \text{lip}\left(\tilde{D}\left(P(L), C_w\right)\right)}{\sum_{L' \in \mathcal{L}} \text{lip}\left(\tilde{D}\left(P(L'), C_w\right)\right)} \leq \sqrt{2^r} \cdot s'(P(L)), \tag{47}
$$

where (45) is by Definition 3.1, (46) is by (44), the left hand side of (47) is by the definition of $\tilde{D}$, and the right hand side of (47) is by Corollary C.1. Also the two factor in the size of $C_w$ is since each line represented by to points with different colors (we may avoid this by modifying the nations in Section 3.1 but we leave the proof for future version). Finally,

$$
s(L) = \sqrt{2^r} \cdot s'(P(L)) \geq S_{\mathcal{L}, k}(L),
$$

where the left hand side is by Algorithm 2, and the right hand side is by (47). □

## C.3 Correctness of Algorithm 3

**Lemma C.5** (restatement of Lemma 3.4). *Let $\mathcal{L}$ be an $(n, m)$-set of lines in $\mathbb{R}^d$, and $k \geq 1$ be an integer. Let $\left(\mathcal{L}^{m+1}, \mathcal{B}^m\right)$ be the output of a call to* LS-DENSE$(\mathcal{L}, k)$*; see Algorithm 3. Then, for every $L \in \mathcal{L}^{m+1}$, we have $S_{\mathcal{L},k}(L) \in O(k) \cdot \left(\frac{1}{|\mathcal{L}^{m+1}|}\right)$.*

The proof of this lemma is inspired by the proof of Lemma 4.1 in [26]. The proof uses the following pair of lemmas. Lemma B.1 [26] whose assumptions hold also for sets of lines, and a generalization of Lemma B.2 [26] for parallel lines.

**Lemma C.6** (Lemma B.1 in [26]). *Let $k \geq 1$ be an integer. Let $A, B$ be a pair of sets of lines in $\mathbb{R}^d$, and $C \subseteq \mathbb{R}^d$ be a set of $|C| = k$ points. If $\tilde{D}(A \cup B, C) \neq \tilde{D}(B, C)$ then $\tilde{D}(A \cup B, C) = \tilde{D}(A, C)$.*

**Lemma C.7** (Generalization of Lemma B.2 in [26]). *Let $A$ be a finite set of lines in $\mathbb{R}^d$, let $\ell \in A$ and $\ell'$ be a line in $\mathbb{R}^d$ that is parallel to $\ell$. Let $B = (A \setminus \{\ell\}) \cup \{\ell'\}$. Then, for every $C \subseteq \mathbb{R}^d$, we have*

$$\tilde{D}(A, C) \leq \rho\left(\tilde{D}(B, C) + \tilde{D}(\ell, \ell')\right).$$

*Proof.* By definition, we have

$$
\begin{aligned}
\tilde{D}(A, C) &= \min\left\{\tilde{D}(\ell, C), \tilde{D}(A \setminus \{\ell\}, C)\right\} \\
&\leq \min\left\{\rho\left(\tilde{D}(\ell, \ell') + \tilde{D}(\ell', C)\right), \tilde{D}(A \setminus \{\ell\}, C)\right\} && (48) \\
&\leq \min\left\{\rho\left(\tilde{D}(\ell, \ell') + \tilde{D}(\ell', C)\right), \rho\left(\tilde{D}(A \setminus \{\ell\}, C) + \tilde{D}(\ell, \ell')\right)\right\} \\
&\leq \rho \min\left\{\tilde{D}(\ell', C), \tilde{D}(A \setminus \{\ell\}, C)\right\} + \rho\tilde{D}(\ell, \ell') \\
&= \rho(\tilde{D}(B, C) + \tilde{D}(\ell, \ell')), && (49)
\end{aligned}
$$

where (48) holds since the distance from a line to a parallel line is the same from every point on the line, hence the weak triangle holds by Definition 2.1, and (49) is by the definition of $B$. $\square$

*Proof of Lemma 3.4.* In what follows, we use the variables and notations from Algorithm 3. Put $L \in \mathcal{L}^{m+1}$, $i \in [m]$, and consider the $i$th iteration of the "for" loop at Line 3 of Algorithm 3. Let $C \subseteq \mathbb{R}^d$ be a set of $|C| = k$ centers such that $\tilde{D}(L, C) > 0$ and $\tilde{D}(T(L, \mathcal{B}^m), C) > 0$. Let

$$\widehat{\mathcal{L}}^{i-1} := \left\{Q \in \mathcal{L}^{i-1} \,\middle|\, \tilde{D}(T(Q, \mathcal{B}^{i-1}), C) = \tilde{D}(\text{proj}(Q, \mathcal{B}^{i-1}), C)\right\}$$

be the union of sets $Q \in \mathcal{L}^{i-1}$ whose closest projected line on $\mathcal{B}^{i-1}$ to the query $C$ is among the lines that are translated to the points of $\mathcal{B}^{i-1}$. Firstly, we prove that

$$\frac{\tilde{D}(T(L, \mathcal{B}^{i-1}), C)}{\sum_{Q \in \mathcal{L}^{i-1}} \tilde{D}(T(Q, \mathcal{B}^{i-1}), C)} \leq 5\rho^2 \frac{\tilde{D}(T(L, \mathcal{B}^i), C)}{\sum_{Q \in \mathcal{L}^i} \tilde{D}(T(Q, \mathcal{B}^i), C)} + \frac{4\rho}{|\mathcal{L}^i|} \qquad (50)$$

via the following case analysis: **(i)** $\left|\widehat{\mathcal{L}}^{i-1}\right| \geq \frac{|\mathcal{L}^{i-1}|}{2}$, i.e., more than half of the sets satisfy that their closest line to $C$ is amongst their translated lines onto $\mathcal{B}^{i-1}$, and **(ii)** Otherwise, i.e., $\left|\widehat{\mathcal{L}}^{i-1}\right| < \frac{|\mathcal{L}^{i-1}|}{2}$.

**Proof for Case (i):** $\left|\widehat{\mathcal{L}}^{i-1}\right| \geq \frac{|\mathcal{L}^{i-1}|}{2}$. By Line 5 in Algorithm 3, we have

$$\mathcal{L}^i \subseteq \mathcal{L}^{i-1} \subseteq \cdots \subseteq \mathcal{L}^0 = \mathcal{L}. \qquad (51)$$

Therefore,

$$
\sum_{Q \in \mathcal{L}^{i-1}} \tilde{D}(T(Q, \mathcal{B}^{i-1}), C) \geq \sum_{Q \in \widehat{\mathcal{L}}^{i-1}} \tilde{D}(T(Q, \mathcal{B}^{i-1}), C) \qquad (52)
$$

$$
= \sum_{Q \in \widehat{\mathcal{L}}^{i-1}} \tilde{D}(\text{proj}(Q, \mathcal{B}^{i-1}), C), \qquad (53)
$$

where (52) holds since $\widehat{\mathcal{L}}^{i-1} \subseteq \mathcal{L}^{i-1}$, and (53) is by the definition of $\widehat{\mathcal{L}}^{i-1}$. This proves (50) for Case (i) as

$$
\frac{\tilde{D}(\mathrm{T}(L,\mathcal{B}^{i-1}),C)}{\sum_{Q\in\mathcal{L}^{i-1}}\tilde{D}(\mathrm{T}(Q,\mathcal{B}^{i-1}),C)} \leq \frac{\tilde{D}(\mathrm{proj}(L,\mathcal{B}^{i-1}),C)}{\sum_{Q\in\mathcal{L}^{i-1}}\tilde{D}(\mathrm{T}(Q,\mathcal{B}^{i-1}),C)}
$$

$$
\leq \frac{\tilde{D}(\mathrm{proj}(L,\mathcal{B}^{i-1}),C)}{\sum_{Q\in\widehat{\mathcal{L}}^{i-1}}\tilde{D}(\mathrm{proj}(Q,\mathcal{B}^{i-1}),C)} \tag{54}
$$

$$
\leq \frac{\tilde{D}(\mathrm{proj}(L,\mathcal{B}^{i-1}),C)}{\sum_{Q\in\widehat{\mathcal{L}}^m}\tilde{D}(\mathrm{proj}(Q,\mathcal{B}^m),C)}, \tag{55}
$$

where the first inequality holds since $\mathrm{T}(L,\mathcal{B}^{i-1}) \supseteq \mathrm{proj}(L,\mathcal{B}^{i-1})$ by Definition 2.7, the second inequality is by (53), and the third is a simple corollary from combining Claim C.2, and Lemma C.4.

**Proof for Case (ii):** $\left|\widehat{\mathcal{L}}^{i-1}\right| < \frac{|\mathcal{L}^{i-1}|}{2}$. Let $\gamma = 1/(2k)$. Let $b^i$, $\mathcal{L}^i$, and $\bar{\mathcal{L}}^{i-1}$ be as defined in Lines 4, and 5, 7, respectively. Identify $\mathcal{B}^{i-1} = \left\{b^1,\ldots,b^{i-1}\right\}$ if $i \geq 2$, and $\mathcal{B}^{i-1} = \emptyset$ if $i = 1$. For every $Q \in \mathcal{L}^{i-1}$, substituting $A = \overline{\mathrm{proj}}(Q,\mathcal{B}^{i-1})$ and $B = \mathrm{proj}(Q,\mathcal{B}^{i-1})$ in Lemma C.6 yields

$$
\left\{Q \in \mathcal{L}^{i-1}\big|\tilde{D}\big(\mathrm{T}(Q,\mathcal{B}^{i-1}),C\big) \neq \tilde{D}(\mathrm{proj}(Q,\mathcal{B}^{i-1}),C)\right\}
$$
$$
\subseteq \left\{Q \in \mathcal{L}^{i-1}\big|\tilde{D}\big(\mathrm{T}(Q,\mathcal{B}^{i-1}),C\big) = \tilde{D}(\overline{\mathrm{proj}}(Q,\mathcal{B}^{i-1}),C)\right\}. \tag{56}
$$

Hence,

$$
\left|\left\{Q \in \mathcal{L}^{i-1} \mid \tilde{D}\left(\mathrm{T}(Q,\mathcal{B}^{i-1}),C\right) = \tilde{D}\left(\overline{\mathrm{proj}}(Q,\mathcal{B}^{i-1}),C\right)\right\}\right|
$$
$$
\geq \left|\left\{Q \in \mathcal{L}^{i-1} \mid \tilde{D}\left(\mathrm{T}(Q,\mathcal{B}^{i-1}),C\right) \neq \tilde{D}(\mathrm{proj}(Q,\mathcal{B}^{i-1}),C)\right\}\right| \tag{57}
$$
$$
= \left|\mathcal{L}^{i-1} \setminus \widehat{\mathcal{L}}^{i-1}\right| \geq \frac{|\mathcal{L}^{i-1}|}{2}, \tag{58}
$$

where(57) is by (56), the equality in (58) is by the definitions of $\mathcal{L}^{i-1}$ and $\widehat{\mathcal{L}}^{i-1}$, and the last inequality is by the assumption of Case (ii).

Recall that by Line 7 of Algorithm 3,

$$
\bar{\mathcal{L}}^{i-1} = \left\{\overline{\mathrm{proj}}(Q,\mathcal{B}^{i-1})\big|Q \in \mathcal{L}^{i-1}\right\},
$$

and define

$$
Z = \left\{Q \in \mathcal{L}^{i-1}\big|\overline{\mathrm{proj}}(Q,\mathcal{B}^{i-1}) \in \mathrm{closest}\left(\bar{\mathcal{L}}^{i-1},C,\frac{1}{2}\right)\right\}.
$$

Since $Z$ contains the $|Z| \leq \left\lceil\frac{|\mathcal{L}^{i-1}|}{2}\right\rceil$ sets $Q \in \mathcal{L}^{i-1}$ with the smallest distance $\tilde{D}(\overline{\mathrm{proj}}(Q,\mathcal{B}^{i-1}),C)$, for any set $Z' \subseteq \mathcal{L}^{i-1}$ such that $|Z'| \geq \frac{|\mathcal{L}^{i-1}|}{2}$, we have

$$
\sum_{Q\in Z}\tilde{D}(\overline{\mathrm{proj}}(Q,\mathcal{B}^{i-1}),C) \leq \sum_{Q\in Z'}\tilde{D}(\overline{\mathrm{proj}}(Q,\mathcal{B}^{i-1}),C). \tag{59}
$$

By the assumption of Case (ii),

$$
\left|\mathcal{L}^{i-1} \setminus \widehat{\mathcal{L}}^{i-1}\right| \geq \frac{|\mathcal{L}^{i-1}|}{2}, \tag{60}
$$

and by the definition of $Z$, we have

$$
\left\{\overline{\mathrm{proj}}(Q,\mathcal{B}^{i-1})\big|Q \in Z\right\} = \mathrm{closest}\left(\bar{\mathcal{L}}^{i-1},C,\frac{1}{2}\right). \tag{61}
$$

Therefore,

$$
\sum_{Q\in\mathrm{closest}(\bar{\mathcal{L}}^{i-1},C,1/2)}\tilde{D}(Q,C) = \sum_{Q\in Z}\tilde{D}(\overline{\mathrm{proj}}(Q,\mathcal{B}^{i-1}),C)
$$
$$
\leq \sum_{Q\in\mathcal{L}^{i-1}\setminus\widehat{\mathcal{L}}^{i-1}}\tilde{D}(\overline{\mathrm{proj}}(Q,\mathcal{B}^{i-1}),C), \tag{62}
$$

where the equality is by (61), and the inequality is by substituting $Z' = \mathcal{L}^{i-1} \setminus \widehat{\mathcal{L}}^{i-1}$ in (59). By the definitions of $\mathcal{L}^{i-1}$ and $\widehat{\mathcal{L}}^{i-1}$, for every $Q \in \mathcal{L}^{i-1} \setminus \widehat{\mathcal{L}}^{i-1}$, we have

$$\tilde{D}(\overline{\text{proj}}(Q, \mathcal{B}^{i-1}), C) = \tilde{D}(\text{T}(Q, \mathcal{B}^{i-1}), C). \tag{63}$$

Let

$$\text{OPT}_i = \min_{C' \subseteq \mathbb{R}^d, |C'|=k} \tilde{D}\left(\text{closest}(\bar{\mathcal{L}}^{i-1}, C', 1/2), C'\right). \tag{64}$$

Hence,

$$\text{OPT}_i \leq \sum_{\bar{Q} \in \text{closest}(\bar{\mathcal{L}}^{i-1}, C, 1/2)} \tilde{D}(\bar{Q}, C) \tag{65}$$

$$\leq \sum_{Q \in \mathcal{L}^{i-1} \setminus \widehat{\mathcal{L}}^{i-1}} \tilde{D}(\overline{\text{proj}}(Q, \mathcal{B}^{i-1}), C) \tag{66}$$

$$= \sum_{Q \in \mathcal{L}^{i-1} \setminus \widehat{\mathcal{L}}^{i-1}} \tilde{D}(\text{T}(Q, \mathcal{B}^{i-1}), C) \tag{67}$$

$$\leq \sum_{Q \in \mathcal{L}^{i-1}} \tilde{D}(\text{T}(Q, \mathcal{B}^{i-1}), C), \tag{68}$$

where (65) holds by the definition of $\text{OPT}_i$, (66) is by (62), (67) is by (63), and (68) holds since $\mathcal{L}^{i-1} \setminus \widehat{\mathcal{L}}^{i-1} \subseteq \mathcal{L}^{i-1}$.

Recall that $\mathcal{B}^m = (b_1, \dots, b_m)$ is an ordered set. Denote the closest line to $b_1$ in $L$ by $\ell_1$, i.e., $\ell_1 \in \arg\min_{\ell \in L} \tilde{D}(\ell, b_1)$. For every $j \in [m-1]$, recursively define $\ell_{j+1}$ to be the line that is closest to $b_i$ over every line in $L \setminus \{\ell_1, \dots, \ell_j\}$, i.e.,

$$\ell_{j+1} \in \arg\min_{\ell \in L \setminus \{\ell_1, \dots, \ell_i\}} \tilde{D}(\ell, b_{j+1}). \tag{69}$$

Hence, for every $j \in [m]$, we have

$$\tilde{D}\left(\overline{\text{proj}}(L, \mathcal{B}^{j-1}), b_j\right) = \tilde{D}(\ell_j, b_j). \tag{70}$$

Since $L \in \mathcal{L}^{m+1} \subseteq \mathcal{L}^i$ and $\gamma = \frac{1}{2k}$, by Line 5 of Algorithm 3 we have

$$\overline{\text{proj}}(L, \mathcal{B}^{i-1}) \in \text{closest}\left(\bar{\mathcal{L}}^{i-1}, \{b^i\}, \frac{(1-\tau)\gamma}{2}\right). \tag{71}$$

By the Pigeonhole Principle, the largest cluster in every set $C'$ of $k$ centers contains at least $\frac{|\bar{\mathcal{L}}^{i-1}|}{k} \leq \frac{|\bar{\mathcal{L}}^{i-1}|}{2k} = \gamma|\bar{\mathcal{L}}^{i-1}|$ sets. Since, by Line 4 of Algorithm 3, $b_i$ is a $(\gamma, \tau, 4)$-median, we have

$$\sum_{Q \in \text{closest}(\bar{\mathcal{L}}^{i-1}, \{b_i\}, (1-\tau)\gamma)} \tilde{D}(Q, b_i) \leq 4 \min_{b \in \mathbb{R}^d} \sum_{Q \in \text{closest}(\bar{\mathcal{L}}^{i-1}, \{b\}, \gamma)} \tilde{D}(Q, b) \leq 4 \cdot \text{OPT}_i. \tag{72}$$

Therefore,

$$\tilde{D}(\ell_i, b_i) = \tilde{D}\left(\overline{\text{proj}}(L, \mathcal{B}^{i-1}), b_i\right) \tag{73}$$

$$\leq 4 \cdot \frac{\sum_{Q \in \text{closest}(\bar{\mathcal{L}}^{i-1}, \{b_i\}, (1-\tau)\gamma)} \tilde{D}(Q, b_i)}{(1-\tau)\gamma|\bar{\mathcal{L}}^{i-1}|} \tag{74}$$

$$\leq 4 \cdot \frac{\sum_{Q \in \text{closest}(\bar{\mathcal{L}}^{i-1}, \{b_i\}, (1-\tau)\gamma)} \tilde{D}(Q, b_i)}{|\mathcal{L}^i|} \tag{75}$$

$$\leq \frac{8\text{OPT}_i}{|\mathcal{L}^i|}, \tag{76}$$

where (73) is by (70), (74) is by combining Markov's inequality with (71), (75) follows since $|\mathcal{L}^i| = \left\lceil \frac{(1-\tau)\gamma}{2}|\mathcal{L}^{i-1}| \right\rceil = \left\lceil \frac{(1-\tau)\gamma}{2}|\bar{\mathcal{L}}^{i-1}| \right\rceil \leq (1-\tau)\gamma|\mathcal{L}^{i-1}|$, and (76) is by (72).

Now, since
$$\mathrm{T}(L,\mathcal{B}^i) = \left(\mathrm{T}(L,\mathcal{B}^{i-1}) \setminus \{\ell_i\}\right) \cup \{\mathrm{T}(\ell_i, b_i)\},$$
i.e., the sets $\mathrm{T}(L,\mathcal{B}^{i-1})$ and $\mathrm{T}(L,\mathcal{B}^i)$ differ only one line, and by Definition 2.7 the line $\mathrm{T}(\ell_i, b_i)$ is parallel to $\ell_i$. Thus, by substituting $A = \mathrm{T}(L,\mathcal{B}^{i-1})$, $B = \mathrm{T}(L,\mathcal{B}^i)$, and $\ell = \mathrm{T}(\ell_i, b_i)$ in Lemma C.7, we obtain
$$\tilde{D}(\mathrm{T}(L,\mathcal{B}^{i-1}),C) \leq \rho\tilde{D}(\mathrm{T}(P,\mathcal{B}^i),C) + \rho\tilde{D}(\ell_i, \mathrm{T}(\ell_i, b_i)). \tag{77}$$
Dividing both sides of (77) by $\sum_{Q\in\mathcal{L}^{i-1}} \tilde{D}(\mathrm{T}(Q,\mathcal{B}^{i-1}),C)$ yields

$$\frac{\tilde{D}(\mathrm{T}(L,\mathcal{B}^{i-1}),C)}{\sum_{Q\in\mathcal{L}^{i-1}}\tilde{D}(\mathrm{T}(Q,\mathcal{B}^{i-1}),C)} \leq \frac{\rho\tilde{D}(\mathrm{T}(L,\mathcal{B}^i),C)}{\sum_{Q\in\mathcal{L}^{i-1}}\tilde{D}(\mathrm{T}(Q,\mathcal{B}^{i-1}),C)} + \frac{\rho\tilde{D}(\ell_i, \mathrm{T}(\ell_i, b_i))}{\sum_{Q\in\mathcal{L}^{i-1}}\tilde{D}(\mathrm{T}(Q,\mathcal{B}^{i-1}),C)}. \tag{78}$$

The rightmost term in (78) can then be bounded by

$$\frac{\rho\tilde{D}(\ell_i, \mathrm{T}(\ell_i, b_i))}{\sum_{Q\in\mathcal{L}^{i-1}}\tilde{D}(\mathrm{T}(Q,\mathcal{B}^{i-1}),C)} \leq \frac{\rho\tilde{D}(\ell_i, \mathrm{T}(\ell_i, b_i))}{\mathrm{OPT}_i} \tag{79}$$

$$= \frac{\rho\tilde{D}(\ell_i, b_i)}{\mathrm{OPT}_i} \tag{80}$$

$$\leq \frac{8\rho\mathrm{OPT}_i}{|\mathcal{L}^i|\,\mathrm{OPT}_i} = \frac{8\rho}{|\mathcal{L}^i|}, \tag{81}$$

where (79) is by (68), and the inequality in (81) holds by (76).

We now bound the middle term of (78). Similarly to (69), for every $Q \in \mathcal{L}^i$ identify $Q = \{q_1,\ldots,q_m\}$. We have,

$$\sum_{Q\in\mathcal{L}^i}\tilde{D}(\mathrm{T}(Q,\mathcal{B}^i),C) \leq \rho\sum_{Q\in\mathcal{L}^i}\tilde{D}(\mathrm{T}(Q,\mathcal{B}^{i-1}),C) + \rho\sum_{Q\in\mathcal{L}^i}\tilde{D}(q_i, \mathrm{T}(q_i, b_i)) \tag{82}$$

$$\leq \rho\sum_{Q\in\mathcal{L}^i}\tilde{D}(\mathrm{T}(Q,\mathcal{B}^{i-1}),C) + \rho\left|\mathcal{L}^i\right|\frac{4\mathrm{OPT}_i}{|\mathcal{L}^i|} \tag{83}$$

$$\leq \rho\sum_{Q\in\mathcal{L}^{i-1}}\tilde{D}(\mathrm{T}(Q,\mathcal{B}^{i-1}),C) + 4\rho\mathrm{OPT}_i \tag{84}$$

$$\leq (5\rho)\sum_{Q\in\mathcal{L}^{i-1}}\tilde{D}(\mathrm{T}(Q,\mathcal{B}^{i-1}),C), \tag{85}$$

where (82) holds by summing (77) over every $Q \in \mathcal{L}^i$, (83) holds since $b_i$ is robst midian for $\bar{\mathcal{L}}^{i-1}$, (84) holds since $\mathcal{L}^i \subseteq \mathcal{L}^{i-1}$ by (51), and (85) is by (68). By (85), the middle term of (78) is bounded by

$$\frac{\rho\tilde{D}(\mathrm{T}(L,\mathcal{B}^i),C)}{\sum_{Q\in\mathcal{L}^{i-1}}\tilde{D}(\mathrm{T}(Q,\mathcal{B}^{i-1}),C)} \leq \frac{5\rho^2\tilde{D}(\mathrm{T}(L,\mathcal{B}^i),C)}{\sum_{Q\in\mathcal{L}^i}\tilde{D}(\mathrm{T}(Q,\mathcal{B}^i),C)}. \tag{86}$$

Combining (78), (81) and (86) yields (50) as

$$\frac{\tilde{D}(\mathrm{T}(L,\mathcal{B}^{i-1}),C)}{\sum_{Q\in\mathcal{L}^{i-1}}\tilde{D}(\mathrm{T}(Q,\mathcal{B}^{i-1}),C)} \leq \frac{5\rho^2\tilde{D}(\mathrm{T}(L,\mathcal{B}^i),C)}{\sum_{Q\in\mathcal{L}^i}\tilde{D}(\mathrm{T}(Q,\mathcal{B}^i),C)} + \frac{8\rho}{|\mathcal{L}^i|}.$$

**Wrapping all together.** We can now apply (50) recursively over every $i \in [m]$ to obtain

$$\frac{\tilde{D}(L,C)}{\sum_{Q\in\mathcal{L}}\tilde{D}(Q,C)} = \frac{\tilde{D}(\mathrm{T}(L,\mathcal{B}^0),C)}{\sum_{Q\in\mathcal{L}^0}\tilde{D}(\mathrm{T}(Q,\mathcal{B}^0),C)}$$

$$\leq (5\rho^2)^m\frac{\tilde{D}(\mathrm{T}(L,\mathcal{B}^m),C)}{\sum_{Q\in\mathcal{L}^m}\tilde{D}(\mathrm{T}(Q,\mathcal{B}^m),C)} + 4\rho\sum_{i=1}^{m}\frac{(5\rho^2)^{i-1}}{|\mathcal{L}^i|}. \tag{87}$$

Furthermore, for every $L' \in \mathcal{L}^{m+1}$ by Lines 9, and 10 of Algorithm 3 and by combining Lemma 3.3 and Lemma C.4 we get

$$\frac{\tilde{D}(\mathrm{T}(L',\mathcal{B}^m),C)}{\sum\limits_{Q\in\mathcal{L}^m}\tilde{D}(\mathrm{T}(Q,\mathcal{B}^m),C)} \leq \frac{2\sqrt{2}mk}{|\mathcal{L}^{m+1}|}. \tag{88}$$

Lemma 3.4 now holds as

$$S_{\mathcal{L},k} = \frac{\tilde{D}(L,C)}{\sum\limits_{Q\in\mathcal{L}}\tilde{D}(Q,C)} \leq \frac{2\sqrt{2}mk(5\rho^2)^m}{|\mathcal{L}^{m+1}|} + 4\rho\sum_{i=1}^{m}\frac{(5\rho^2)^{i-1}}{|\mathcal{L}^i|} \tag{89}$$

$$\leq \frac{2\sqrt{2}mk(5\rho^2)^m}{|\mathcal{L}^{m+1}|} + 4\rho\sum_{i=1}^{m}\frac{(5\rho^2)^{i-1}}{|\mathcal{L}^{m+1}|} \tag{90}$$

$$\leq \frac{2\sqrt{2}mk(5\rho^2)^m}{|\mathcal{L}^{m+1}|} + \frac{4\rho}{|\mathcal{L}^{m+1}|}\cdot\frac{(5\rho^2)^{m-1}-1}{(5\rho^2)-1} \tag{91}$$

$$\leq \frac{2\sqrt{2}mk(5\rho^2)^m}{|\mathcal{L}^{m+1}|} + \frac{4\rho}{|\mathcal{L}^{m+1}|}\cdot(5\rho^2)^m \tag{92}$$

$$\leq \frac{15mk\rho(5\rho^2)^m}{|\mathcal{L}^{m+1}|}, \tag{93}$$

where (89) holds by plugging (88) in (87), (90) holds since $|\mathcal{L}^{m+1}| \leq |\mathcal{L}^i|$ for every $i \in [m]$, the last derivation holds by summing the geometric sequence, and inequalities (92) and (93) hold since $\rho \geq 1$. $\square$

**Overview of Algorithm 6**    Suggested implementation for robust median; See Definition 2.5.

---
**Algorithm 6:** MEDIAN$(\mathcal{P}, k, \delta)$

---
1 **Input:** An $(n, m)$-set $\mathcal{P}$, a positive integer $k \leq 1$, and probability of failure $\delta \in (0, 1)$.
2 **Output:** A point $q \in \mathcal{X}$ that satisfies Lemma C.8
3 $b :=$ a universal constant that can be determined from the proof of Lemma C.8
4 Pick a random sample $\mathcal{S}$ of $|\mathcal{S}| = b \cdot k^2 log(\frac{1}{\delta})$ sets from $\mathcal{P}$
5 $q :=$ a point that minimizes $\sum_{p\in\mathrm{closest}(\mathcal{S},\{q\},(1-\tau)\gamma)}\tilde{D}(p,q)$ over $q \in Q \in \mathcal{S}$
6 **Return** $q$

---

**Lemma C.8** (Lemma 5.1 in [26]). *Let $\mathcal{P}$ be an $(n, m)$-set in $\mathcal{X}$, $k \geq 1$, and $\delta \in (0, 1)$. Let $q \in \mathcal{X}$ be the output of* MEDIAN$(\mathcal{P}, k, \delta)$*; see Algorithm 6. Then, with probability at least $1 - \delta$, $q$ is a $\left(\frac{1}{2k}, \frac{1}{6}, 2\right)$-median for $\mathcal{P}$; see Definition 2.5. Moreover, $q$ can be computed in $O(tb^2k^4\log(\frac{1}{\delta}))$ time, where $t$ is the time it takes to compute $\tilde{D}(P, Q)$ for every pair $P, Q \in \mathcal{P}$.*