# OpenReview forum: "Coreset for Line-Sets Clustering"
_NeurIPS.cc/2022/Conference — NeurIPS 2022 Accept_

### Official Review · Reviewer_KAjB · 2022-07-09

**Rating:** 5
**Confidence:** 5
**Soundness:** 2 fair
**Presentation:** 2 fair
**Contribution:** 2 fair

**Summary:**

This paper investigates the line-sets k-median problem, whose goal is to compute a set of k centers that minimizes the total Euclidean distances from each set of lines to its closest center. The authors propose an algorithm that constructs a small-size coreset for the line-sets k-median problem, based on a novel reduction to a colored point-sets clustering problem and a coreset algorithm for this problem.



**Questions:**

Could the authors explain why the algorithms work? E.g.,
- What is the intuition behind Lemma 3.4?
- How to obtain the coreset size in Theorem 3.5 from Lemma 3.4?

**Limitations:**

Yes

**Strengths And Weaknesses:**

This paper proposes a new problem:  the line-sets k-median problem, a generalization of clustering lines. The proposed coreset algorithm is non-trivial and relies on a reduction to a colored point-sets clustering problem.

My main concerns are the writing and the significance.
- I don't think the line-sets k-median problem is common in practice. The authors provide a discussion on the applications, e.g., computer vision, but somehow artificial. The studied problem is more like a technical problem instead of a practical one.
- The technical part is difficult to read. 1) Def 3.2 appears a little late, while the authors have mentioned ordered-set before. 2) No discussion or explanations on the algorithms, which makes the reading not easy. 3) The main coreset algorithm is in the appendix, which makes me surprised. 4) By the algorithms, I don't understand what the technical difficulty is and why the algorithm is correct. Some intuitions should be given.

------------------------------------------------------------------------------------------------------------------------------------------------------------------------------------------
Thanks for the response. The revised pdf is better in writing. I raise my score accordingly.

---

> ### Author Response · Authors · 2022-08-02
> **Response to Reviewer KAjB**
>
> ## Thank You
> We thank the reviewer for their time and effort in reviewing our paper and greatly appreciate the issues raised.
>
> Below we provide a detailed response to all of the issues raised and we would be more than happy to further engage with the reviewer at any time during the discussion period to clear up the remaining issues. Overall, we hope our response will encourage you to increase your score.
>
> ## Reviewer's Questions
>
> **Question 1:** ``What is the intuition behind Lemma 3.4?''
>
> **Answer:** Following this good question, we added the following explanation to Section~C.1. The goal of the algorithm is to find a small family (set) $\mathcal{P'}$ of $\Theta(n)$ sets that are sufficiently mutually close that they can be replaced with multiple copies of the same set with little to no effect on the cost. In that case, all of their sensitivities would be $\sim\frac{1}{|\mathcal{P'}|}$. The proof of this lemma is by case analysis of two cases. The first case assumes that one of the centers in the query is close to $\mathcal{P'}$. In this case, the sets in $\mathcal{P'}$ are not affecting the cost at all. In the other case, we assume that all of the centers are far from $\mathcal{P'}$, and then we try to show that they are much closer to each other than to the centers relative to the cost of the centers.
>
> **Question 2:** ``How to obtain the coreset's size in Theorem 3.5 from Lemma 3.4?''
>
> **Answer:** The coreset's size is determined by substituting the total sensitivity and VC-dimension in the statement of TheoremB.8. The VC-dimension is bounded in Lemma B.6, and the total sensitivity is bounded in the proof of Theorem 4.1, B.9 in the appendix (previously 3.5 and B.10).
> ## Additional Concerns
>
> **Question 3:** ``Line-sets k-median problem is not so common in practice. The authors provide a discussion on the applications, e.g., computer vision, but somehow artificial.''
>
>  **Answer:** Indeed, problems such as $k$-line means are not so common as the very common $k$-(points)-mean clustering and its versions.
> We believe that the reason is that these problem are much harder and have no practical solution since the applications for missing data/labels or computer vision are straightforward. Our hope is that our paper, and especially the open source code will initiate a change toward this direction.
>
> **Question 4:** ``Definition 3.2 appears a little late, while the authors have mentioned ordered-set before.''
>
> **Answer:** We thank the reviewer for pointing this out. Definition 3.2 (now 2.4) is now on Section 2.
>
> **Question 5:** ``Explain the technical difficulty and why the algorithm is correct. Some intuitions should be given.''
>
>  **Answer:**  The intuition for the algorithms is given in Section~1.5. Following this suggestion, we added Fig.2 from the appendix to support it. In addition, we have extended the algorithms overview; see the next response below.
>
> **Question 6:** `` Add discussions and explanations on the algorithms.''
>
>  **Answer:**  Following this suggestion, we added more details and intuition as to what the algorithm does and why. In the Overviews for all the algorithms that appear before each algorithm.
>
> **Question 7:** ``  The main coreset algorithm is in the appendix''
>
>  **Answer:**  Following your suggestion, we have moved the algorithm to the main section of the paper.
>
> ## Final Remark
> Finally, we thank the reviewer for the comments that helped us further improve the quality of this paper. We hope that we answered the main concerns and convinced the reviewer to raise the score. If not, we would very much appreciate learning what is still missing during the discussion period.

---

> ### Author Response · Authors · 2022-08-08
> **Thanks + Please inform us of any remaining issues.**
>
> Dear reviewer (KAjB),
>
> We thank you for your response and are encouraged by the increased score.\
> We appreciate your feedback and honestly believe it significantly improved the paper. \
> Is there any other concern we may address in the hope of increasing our score?

---

### Official Review · Reviewer_y2Tf · 2022-07-10

**Rating:** 7
**Confidence:** 3
**Soundness:** 4 excellent
**Presentation:** 4 excellent
**Contribution:** 3 good

**Summary:**

This paper proposes the problem of k-clustering for a collection of sets of lines. In this new problem, each “data point” is a set of 1D lines in R^d. The goal is still to find a center set in R^d of k points, such that the sum of “distance” to every data point is minimized, where this “distance” is defined as the minimum distance from the center set C to the nearest line from a line set.

Even though the problem looks artificial at the first glance, it does have natural and important applications. For instance, as pointed by the authors, the line clustering itself might be viewed as clustering points each with a single missing numerical feature (so all possible realizations of that missing coordinate form a line), and a further missing categorical feature results in a set of lines.

The paper focuses on studying the coreset for the line sets clustering problem. As usual, the definition of coreset is still to preserve the cost for every possible center set C \subseteq R^d, up to 0 < \epsilon < 1 relative error. The main result is an efficient coreset construction with size poly(m log n / eps) k^{mk}, where m is the (max) number of lines in each line set.

This result might be viewed as a generalization of previous papers [26, 28], which deal with the case m = 1 or the case of sets of points. To resolve the new challenge of m > 1 lines, an auxiliary problem called clustering with colored sets is considered, where a color corresponds to a line in a line set, and it is used to ensure that only one line that achieves the minimum is assigned to the center set. The overall approach is similar to [26] and [28] which is also based on the sensitivity sampling framework, but the colored version of clustering is a necessary new step, especially used to compute the sensitivity. I find it natural and elegant.

Experiments that evaluate the empirical size-vs-accuracy of the proposed coresets are conducted, compared with a uniform sampling baseline, on both synthetic and real datasets with moderate size (~10^4 points). The results show an error curve that is superior to uniform sampling in the sense that it achieves much better error for smaller coreset size, and both algorithms converge when the number of coreset points reaches ~10^3.


**Questions:**

1. It seems uniform sampling already performs well enough from the experiment – I understand the error curve is slightly better, but I don’t think that matters a lot in practice. Can you justify the absolute advantage of your coreset over uniform sampling? At least I cannot see it from the plot, nor from your interpretation of the result.

2. You mentioned several times the colored clustering problem is tightly related to fair clustering. I can see they seem to be related, but the more concrete/formal relation is not clear to me. In particular, does this give you any result for the setting of fair clustering directly?

Minor comments:

1. Line 211: “Langberg-Feldman” -> “Feldman-Langberg”
2. Line 246: “lth”->’l-th”
3. Line 4 of algorithm 1, it’s better to use \mid in the set notation – currently the spacing is too small.
4. Line 260, “that are both satisfy” -> “that both satisfy”


**Limitations:**

Yes, and it has been discussed in the checklist.

**Strengths And Weaknesses:**

Overall, I think this is a nice generalization of [26] and [28], especially considering the motivation for dealing with the missing value on categorical features. The result is probably nearly the best that one can obtain using current techniques, but this also means that one still does not have a full understanding of the problem, and new techniques and/or lower bound are much appreciated. Even though this paper certainly has theoretical value, the experiment result seems not completely convincing.

The paper is well written and a comparison with previous works seems to be comprehensive.

---

> ### Author Response · Authors · 2022-08-02
> **Response to Reviewer y2Tf**
>
> ## Thank You
>
> We thank the reviewer for appreciating our work and the given encouraging score. In what follows we address the questions raised from the initial review. We hope to continue to improve the paper and answer concerns during the discussion period.
> ## Reviewer's Questions
>
> **Question 1:** ``It seems uniform sampling already performs well enough from the experiment – I understand the error curve is slightly better, but I don’t think that matters a lot in practice. Can you justify the absolute advantage of your coreset over uniform sampling?''}
>
> **Answer:** We thank the reviewer for their question and would like to point out a few things:
> 1. We agree that in some real-life databases the margin is small. In others, the margin between coreset and uniform sampling might increase to infinity. Please keep in mind the coreset has worst-case guarantees, unlike uniform sample. Unlike other papers, we publish the results of the first tests that we found, with no hidden cherry-picking. The strength of the coreset here lies in its reliability. The coreset proved to work even in the worst case in real life you can't measure the quality of the compression.
>   2.  Even in plots where the margin is small, it is possible to see that the variance of the uniform sampling is large while the coreset is very consistent.
>    3. One of the main motivations for using coresets is the support of streaming as well as distributed and dynamic data (including insertions and deletions), as explained e.g. in [10] In this case, the probability that the uniform sample becomes useless in terms of high approximation error increases very fast with the number of updates.
>  4. Lastly, the presented experiments show that are similar to many other published papers. In fact, in the new experiments added according to the reviewer fpBB suggestions, our coreset outperforms existing coresets.
>
> **Question 2:** ``Can you provide formal relation to fair clustering. Does this give you any result for the setting of fair clustering directly?''
>
> **Answer:** This is an interesting question that we did not pursue in this paper. We added it to the future work section. We believe that the answer is "no", but expect that the new suggested techniques would help to either extend or improve existing results.
>
> ## Final Remark
>
> We would like to thank the reviewer again for their encouraging review. We would also like to address any other questions the reviewer may have during the discussion period.

---

> > ### Comment · Reviewer_y2Tf · 2022-08-08
> > **Thanks for the response**
> >
> > I agree that uniform sampling probably won't work well for streaming/distributed/dynamic settings when updates to data need to be handled. But then it would be interesting to see a validation of this. For instance, one might try to see how this uniform sampling works with the standard merge-and-reduce framework in datasets.
> >
> > I also agree with other reviewers that the comparison to other baselines is welcome.

---

> > > ### Author Response · Authors · 2022-08-08
> > > **Thanks**
> > >
> > > Dear reviewer (y2Tf),
> > >
> > > We thank you for your reply.
> > > We still do not have streaming and distributed versions of our coreset but believe it will be ready during the next week or two.
> > > It is based on the code from GitHub/YairMarom/k_lines_means . We again thank the reviewer for the professional and positive review.

---

### Official Review · Reviewer_fpBB · 2022-07-11

**Rating:** 7
**Confidence:** 3
**Soundness:** 3 good
**Presentation:** 2 fair
**Contribution:** 2 fair

**Summary:**

The paper studies the problem of clustering $n$ sets of $m$ lines. In detail, the objective is to compute $k$ means that minimize the squared Euclidean distance to the closest point on the closest line in each of the $n$ sets. This problem is a novel generalization of the $k$-means for lines clustering problem, which has already been studied in the literature.

**Questions:**

My suggestion is to remove the experiments and include more theoretical work. It would be nice to put the reduction to colored points into focus.

**Limitations:**

No potentially negative societal impact in sight.

**Strengths And Weaknesses:**

The paper is well written and the problem is somewhat interesting, though I am not completely convinced on the motivation of studying this problem. The authors motivate it by giving an example with missing values. However, the missing value is arbitrary and the objective function is a chain of minima, which makes me doubt the practical expressiveness of a solution.

The reduction from the problem given $n$ sets of $m$ lines to $n$ sets of $m$-tuples of points (colored points) is interesting (as is the latter problem). Unfortunately, little of the analysis has made it into the main paper.

The general result is obtained by an application of the sensitivity sampling framework, which is a standard in the coreset literature (at least for sum-based clustering objectives).

A minor weakness is that the related work is treated a bit superficially. The coreset size from the $k$-means for lines work is not stated -- it would be interesting to compare their size to the coreset size of the work at hand. Furthermore, the experiments are very basic and only compare to uniform random sampling -- it would be much more interesting to also include here the work of [33] and compare to them for $m=1$. Also it is sometimes not clear whether the authors are aware that projective clustering has already been (extensively) studied.

---

> ### Author Response · Authors · 2022-08-02
> **Response to Reviewer fpBB**
>
> ## Thank You
> We thank the reviewer for the comments and the helpful suggestions. It is not so common to get such detailed and professional reviews. Here we provide answers to the issues and suggestions raised from the initial review. If there are still open questions or more answers we would be glad to answer them during the discussion period.
>
> ## Comparison with [36] (previously 33)
>
> Following the reviewer's suggestion, we add to the existing citation also a brief description and the coreset's size.
>
> **Theoretical bounds on coreset's size.** We emphasize the fact that the result of [36] applied only for m=1. As both coresets are based on sensitivity sampling, comparing the upper bound on the total sensitivity would be equivalent to comparing the coreset sizes.
> | Paper | Total sensitivity |
> |-------|-------------------|
> | ours (assuming $m=1$)   | $k^{k+1}\log(n)$               |
> | coreset for $k$-mean for lines     | $k^{\Omega(k)}\log(n)$              |
>
> We have added this comparison to the new version.
>
>
> **Experimental results.** Following the reviewer's suggestion. In the new revision of the paper, we have added two new experiments for $m=1$ and $k=\{1,2\}$, in which we compare our coreset to both uniform sampling and coreset for $k$-mean for lines. The results are shown in the new revision and also attached below. As expected, the results of both coresets are similar and better than uniform sampling but our coreset outperforms previous work as our bound for the sensitivity is tighter.
>
> ## Additional Concerns
>
> **Question 1:** ``the missing value is arbitrary ..., which makes me doubt the practical expressiveness of a solution.''
>
>  **Answer:** We agree that the main contribution of this paper is the theoretical breakthrough and results. However, we also expect that our code can be extended and applied for real-world systems that have similar problems, such as [26,35] (in the new version). We added this paragraph to the Future Work section and thank the reviewer for emphasizing this.
>
> **Question 2:** ``remove the experiments and include more theoretical work''
>
>  **Answer:** Following Answer 1 above, one of the reasons that we did not send this paper to a pure theoretical CS conference is that we believe that practitioners can also use it for real-world systems. This is also why we spent a lot of time implementing the code and asked our sponsors to make it open.
>
>
> **Question 3:** ``put the reduction to colored points into focus.''
>
>  **Answer:** The reduction is described in Section 1.5, and the technical details are in Section C.2. We have added Fig.2 to Section 1.5. Please let us know if there are any specific details that you felt were missing.
>
> **Question 4:** ``Also it is sometimes not clear whether the authors are aware that projective clustering has already been (extensively) studied''}
>
>  **Answer:** We actually well aware of these results.
> They were the main motivation for this paper in the sense that we wish to generalize projective clustering in two directions:
>
>     (i)  Input $n$ subspaces instead of n points
>     (ii) Input $n$ sets instead of n singletons
> After at least two decades of research in projective clustering, the first results for (i) and (ii) appear only in the recent years, mainly in NeurIPS/ICML and are cited in Section 1. More unique to this paper is the combination of both (i) and (ii). We hope that our paper will boost the research that generalizes projective clustering.
>
> ## Final Remarks
> We thank the reviewer again for their insightful comments, and we are looking forward to continuing the discussion. Hopefully, the reviewer will consider raising their score given that we have added the requested experiments and comparisons and addressed all their feedback.

---

> > ### Comment · Reviewer_fpBB · 2022-08-08
> > **Response to the Authors**
> >
> > Dear authors,
> >
> > thank you for the comprehensive answer! Concerning the related work and the experiments, my concerns have been adequately addressed. I am not completely sold on the meaningfulness of the stated extension on projective clustering. Also, this work does only cover two-dimensional subspaces (?)
> >
> > Nevertheless, I will raise my score!

---

> > > ### Author Response · Authors · 2022-08-08
> > > **Thanks + Additional information about projective clustering**
> > >
> > > Dear reviewer (fpBB),
> > >
> > > We thank you for your response and very much appreciate the increased score.
> > > We are sorry that the answer was not clear and try a more detailed one as follows.
> > >
> > > Almost all the papers in projective clustering assume that the input is a set of points and not a set of higher dimensional subspaces, such as lines.
> > > In addition, very few papers assume that the input is a set of sets and not just a set of points.
> > >
> > > On the contrary, our paper assumes that the input is a set of sets of lines.
> > > Handling this new combined setting was also our main challenge.
> > > We expect that the paper will open the door for generalization to "projective clustering of sets of subspaces" (not only lines) and believe that combining this paper with classic projective results would make this generalization significantly easier for the community.
> > >
> > > We hope that we clear everything. Nevertheless, we will add many more citations to projective clustering in the related work.
> > >
> > > Thank you again for the professional review and the encouraging score.

---

### Author Response · Authors · 2022-08-02
**General Response to all reviewers**

## Thank You

We thank the reviewers for their positive feedback and helpful suggestions. We highly appreciate the effort to provide in-depth reviews that helped us to improve our work.
## Positive feedback
We were glad to hear that reviewers fpBB and y2Tf thought that our work is **``well writen''** and moreover, they mentioned that our work is **``interesting''** and **`` does have natural and important applications''** (respectively). In addition, that reviewer KAjB found our work **``non-trivial''**.

## Detailed Response

Due to the requests of the reviewers, especially the request to add Algorithm 4, the paper is now about 9.5 pages long. We put the new version in the supp. material in the hope that you will accepted it for the (10 pages) camera ready version.

In addition,
We will be addressing reviewer comments in detail below in separate responses. If there are any remaining concerns, please do not hesitate to raise these issues and we would be more than happy to address them in a timely manner.

## Further discussion
Overall, we believe we were able to address the reviewers’ concerns to an extent that will hopefully convince the reviewers to raise their scores.

We look forward to further engaging with you during the up-coming discussion period.

---

### Author Response · Authors · 2022-08-07
**Responses Follow-up**

Dear all reviewers,

First, we thank you again for your initial insightful reviews.\
 We would also like to know whether we addressed your concerns satisfactorily. If some of your concerns were left unresolved, please let us know so we are able to address them during the short time left for the discussion period.

Many thanks,\
The authors.

---

### Author Response · Authors · 2022-08-08
**Summary**

Dear Reviewers, ACs, and SACs,

As the discussion period is coming to an end, we would like to express our deep gratitude for spending the time to review and assess our paper, as well as for providing us with insightful comments throughout the review process. Your feedback has already helped us to improve our paper significantly.

We hope we were able to address all the reviewers’ concerns and issues.

We are here to address any lingering issues and comments at short notice as the review process is coming towards an end.

Thank you,\
The authors

---

### Meta-Review · Area_Chair_XG9Z · 2022-08-25

**Recommendation:** Accept
**Confidence:** Less certain

**Metareview:**

The paper gives new coresets for sets of lines in a high dimensional space. The problem comes from modeling an input data point with 2 missing attributes, one continuous and one discrete, as a set of lines. Previous results only worked with missing continuous attributes. The reviewers appreciate the technical contribution of the paper and it might lead to additional results for data with more general patterns of missing attributes. On the other hand, the contribution seems somewhat limited as the previous works for missing continuous attributes can handle a general number of missing attributes, up to 10-20 whereas the formulation here is very specific, albeit with 1 discrete attribute. More experimental evaluation comparing with previous coreset results in common special case would also strengthen the paper.

**Award:**

No

---

### Decision · Program_Chairs · 2022-09-14

Accept